# DCAF1-based PROTACs with activity against clinically validated targets overcoming intrinsic- and acquired-degrader resistance

Martin Schröder [1,6] ✉, Martin Renatus[1,4,6], Xiaoyou Liang[2], Fabian Meili [2], Thomas Zoller[1], Sandrine Ferrand[1], Francois Gauter[1], Xiaoyan Li[2], Frederic Sigoillot [2], Scott Gleim[2], Therese-Marie Stachyra [1], Jason R. Thomas[2], Damien Begue[1], Maryam Khoshouei[1], Peggy Lefeuvre[1], Rita Andraos-Rey[1], BoYee Chung[2], Renate Ma[2], Benika Pinch[2], Andreas Hofmann [1], Markus Schirle[2], Niko Schmiedeberg[1], Patricia Imbach[1], Delphine Gorses[1], Keith Calkins[1], Beatrice Bauer-Probst[1], Magdalena Maschlej[1], Matt Niederst[2], Rob Maher[2], Martin Henault[2], John Alford[2], Erik Ahrne[1], Luca Tordella[1], Greg Hollingworth[1], Nicolas H. Thomä [3,5], Anna Vulpetti [1], Thomas Radimerski[1,4], Philipp Holzer[1], Seth Carbonneau[2] & Claudio R. Thoma[2,4] ✉

Targeted protein degradation (TPD) mediates protein level through small molecule induced redirection of E3 ligases to ubiquitinate neo-substrates and mark them for proteasomal degradation. TPD has recently emerged as a key modality in drug discovery. So far only a few ligases have been utilized for TPD. Interestingly, the workhorse ligase CRBN has been observed to be downregulated in settings of resistance to immunomodulatory inhibitory drugs (IMiDs). Here we show that the essential E3 ligase receptor DCAF1 can be harnessed for TPD utilizing a selective, non-covalent DCAF1 binder. We confirm that this binder can be functionalized into an efficient DCAF1-BRD9 PROTAC. Chemical and genetic rescue experiments validate specific degradation via the CRL4$^{DCAF1}$ E3 ligase. Additionally, a dasatinib-based DCAF1 PROTAC successfully degrades cytosolic and membrane-bound tyrosine kinases. A potent and selective DCAF1-BTK-PROTAC (DBt-10) degrades BTK in cells with acquired resistance to CRBN-BTK-PROTACs while the DCAF1-BRD9 PROTAC (DBr-1) provides an alternative strategy to tackle intrinsic resistance to VHL-degrader, highlighting DCAF1-PROTACS as a promising strategy to overcome ligase mediated resistance in clinical settings.

Targeted protein degradation (TPD) eliminates proteins through the ubiquitin-proteasome pathway, the cellular disposal system. TPD has been actively pursued for more than a decade as a new therapeutic modality with multiple small molecule-based degraders currently in clinical development[1]. Rather than inhibiting proteins, TPD mediates

protein elimination through attachment of ubiquitin chains marking them for degradation. On a molecular level, small molecule-mediated degradation can be achieved by two classes of molecules. The first class encompasses bivalent molecules, also known as proteolysis targeting chimeras (PROTACs), which induce proximity between an E3

ligase and the protein of interest (POI) through two independent binding events—this is mediated by two separate binding moieties connected by a linker[2,3]. The second class includes monovalent molecules, known as molecular glues, which bind to and reshape the surface of an E3 ligase receptor, enabling a novel and stabilizing protein-protein interaction with the POI.

The identification of E3 ligase binders provides the chemical handle to exploit E3 ligases for TPD approaches. Even though there are ~600 E3 ligases encoded in the human genome, so far only a handful have been successfully engaged with non-covalent binders for TPD. CRBN is the most widely used E3 which is hijacked by IMiD derivatives. The second most described ligase for TPD, the von Hippel-Lindau protein (VHL)[4], is less frequently used in PROTACs or in clinical development candidates compared to CRBN[1,5]. Other than the CRBN and VHL E3 ligase binders that can mediate TPD at low and sub nM potencies[5–7], there are few additional non-covalent ligase binders that have matched the potencies of those E3 handles in TPD applications[8–10]. Other promising non-covalent E3 ligase binders with characterized but less potent PROTACs for TPD of neo-substrates have been described for KEAP1[11], MDM2[12], and IAP[13]. In addition, multiple chemical proteomics approaches using electrophiles have enabled the discovery of additional tractable ligases for TPD[14] such as DDB1 and CUL4 associated factor 16 and 11[15–17], RNF4[18] and more recently DCAF1 (also known as Vpr binding protein VprBP)[19], although this latter degrader prototype only works in conditions of DCAF1 ectopic expression. Covalent ligands of E3 ligases have the limitation of reducing the catalytic nature of PROTAC molecules because they are dependent on the half-life of the E3 receptor domains. Furthermore, the covalent occupancy of the E3 ligase receptor might block degradation of natural substrates and as such may be more prone to on-target toxicity. The irreversible nature of covalent E3 binders could also complicate the generation of highly specific ligase binders due to reduced selectivity for the respective ligase.

The most widely used IMiDs for TPD are thalidomide and its next-generation derivatives lenalidomide and pomalidomide, which have been approved as 1st line treatment for multiple myeloma (MM). MM is a plasma cell malignancy that is dependent on IKZF transcription factors (reviewed in ref. [20]). Of note, thalidomide and its analog lenalidomide were approved by the FDA for the treatment of MM many years before their TPD MoA was described. Unfortunately, a recurring problem in IMiD-treated MM patients is emergence of resistance. About 30% of the IMiD-refractory patients bear alterations in CRBN levels, with copy loss in most cases, and lower CRBN levels result in reduced IMiD-mediated IKZF degradation. CRBN is a non-essential gene based on genome-wide CRISPR KO studies[21–23], which raises the question if loss of CRBN expression, or mutations in CRBN might be a liability for CRBN-targeting PROTACs? Therefore, targeting essential E3 ligases represents a potential strategy to circumvent or at least delay the emergence of therapy resistance[24].

Here, we describe the discovery of non-covalent PROTACs based on a specific binder to the E3 ligase receptor DCAF1[25]. DCAF1 is an essential WD40 repeat (WDR) domain containing E3 ligase receptor of the Cullin RING ligase (CRL) 4 subfamily[26]. Using two different PROTAC prototypes, one directed towards the nuclear protein BRD9 and the other towards tyrosine kinases, we demonstrate utility of DCAF1 for targeted protein degradation in cellular models, under physiological conditions. Using a selective Bruton's tyrosine kinase (BTK) inhibitor, we extend our studies by building a DCAF1-based degrader of BTK. CRBN-based BTK degraders have recently entered the clinic and might suffer from the same resistance mechanisms reported for other degraders using CRBN. In this work, we show that DCAF1 can be harnessed for BTK degradation in preclinical settings, including those that have developed resistance to CRBN PROTACs, providing the basis for developing next-generation BTK-targeting therapeutics.

## Results

### Discovery and characterization of a selective DCAF1 E3 ligase receptor binder

Successful discovery of ligands for WDR domain proteins such as EED[27] and WDR5[28,29] shows that WDR domains can be targeted (reviewed in[30]). Additionally, a recent publication described reversible ligands binding to the WD40 domain of DCAF1, highlighting further the potential of developing non-covalent DCAF1-based PROTACs[31]. Amongst the ~600 E3 ligases, we focused on the 48 E3 ligases that contain WDR domains as putative recognition motifs (Supplementary Data File 1). In addition, as a rationale to have potentially a higher bar for emerging resistance mechanisms on the ligase side, we analyzed this subset of 48 E3 receptors for essentiality by extracting the Demeter2 dependency scores from the DepMAP database[23,32]. Interestingly, the mainly nuclear expressed E3 ligase receptor, DCAF1, a part of the CRL[DCAF1] and EDVP complex[33,34], showed similar tumor essentiality in the test set to DDB1 (Fig. 1a). Since both proteins are direct interacting partners in the CRL-DCAF1 complex, the essentiality of DDB1 might even be linked to the cellular presence or absence of DCAF1. The DCAF1-DDB1 complex serves as a CRL4 receptor that possesses an inactive tetrameric state, which upon neddylation, transitions into an active dimer[35]. The DCAF1 WD40 domain, as mentioned above[31], showed promising degradation results targeted by a covalent PROTAC, however, only at artificially high DCAF1 expression levels[19].

DCAF1 is frequently hijacked by Vpr and Vpx, virion-associated proteins encoded by lentiviruses, such as the human immunodeficiency virus[26]. Mechanistically, they act as a protein glue to recruit cellular restriction factors to DCAF1 for ubiquitination and subsequent degradation, setting a precedent for de novo substrate degradation via DCAF1[36]. Furthermore, DCAF1 has been shown to be a preferred receptor in Cullin4 ligase assemblies and ranks as the second most abundant CRL4 receptor in CRL4 E3 ligases behind CRBN[37]. In addition, various DCAF1 substrates have been proposed such as MCM10[38] or FOXM1[39].

A potential lack of genetic dependency on CRBN in cancer cells as exemplified by the Demeter2 score might explain rapid emergence of resistance in patients (Fig. 1b). Therefore, we hypothesized that the essentiality of DCAF1 might provide a higher bar for the occurrence of resistance. Furthermore, comparison of genetic dependency scores between DCAF1 and other ligase receptors that have been liganded for TPD approaches highlighted the potential of DCAF1 as an essential gene with respect to its Demeter2 score (Fig. 1c).

Our DCAF1 binder discovery strategy and medicinal chemistry campaign are described in an accompanying manuscript[25]. In summary, WDR binders from a previous hit-discovery campaign for EED[27] have been screened and optimized using NMR, computer-aided drug design, biophysics and structure–activity relationship approaches. This resulted in the discovery of a scaffold that occupies the DCAF1 WDR donut-hole pocket. Here, we further characterized the potent DCAF1 binder (**13**), which binds to DCAF1 with a $K_d$ by SPR and $IC_{50}$ by TR-FRET of <50 nM (Fig. 1d–g). X-ray structure of **13** bound to DCAF1 confirmed the same binding mode of the previously described DCAF1 donut-hole binders[25] and supported to explore the piperazine as a potential site for an exit vector (Fig. 1h)[40]. As potential control compound, key for robust validation as highlighted by ref. [40], we synthesized (**13-N**) (Fig. 1d). This di-methylated analog of the primary amine showed over 100× fold weaker binding affinity in both SPR and TR-FRET assays (Fig. 1e–g, Supplementary Fig. 1a). This reduced affinity is ascribed to a suboptimal positioning of the ternary amine with consequent loss of the hydrogen bonds with D1356 in which the primary amine is involved. Extension of compound (**13**) from the piperazine by PEG6 chain(**15**) (Supplementary Fig. 1b), did not alter binding to DCAF1 as confirmed by X-ray crystallography (Supplementary Fig. 1c) and placing the exit vector in the potential zone of ubiquitination (Supplementary Fig. 1d), thus allowing functionalization

 **2**

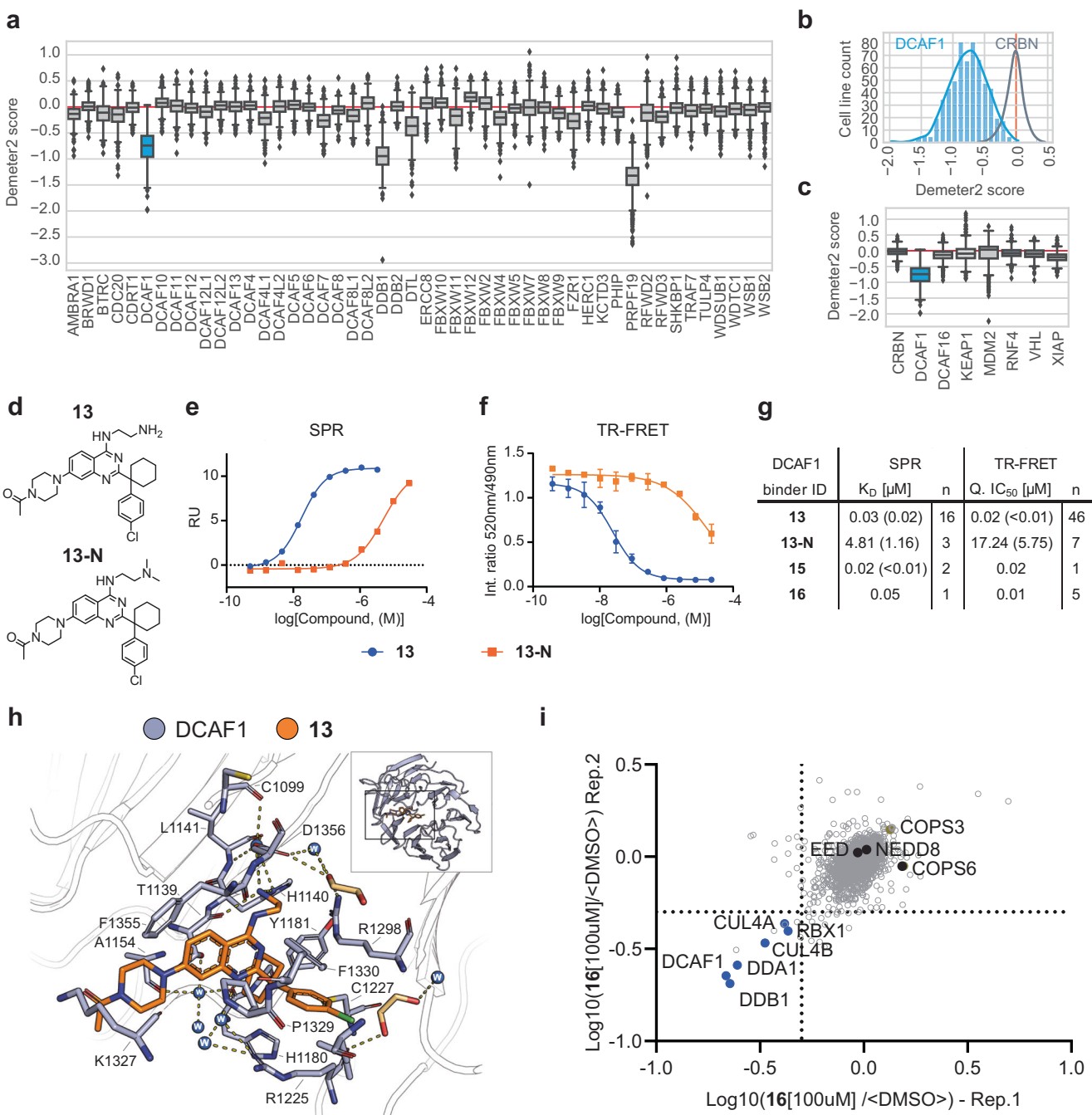

**Fig. 1 | DCAF1 ligand characterization for targeted protein degradation. a** RNAi gene effect score (Demeter2) extracted for WDR domain containing E3 ligases. Sample sizes represented by these scores range from a maximum 711 (DDB1) to a minimum 343 (FBXW12) cell line counts out of a maximum possible 712 cell lines. The complete list of these gene-sample counts is summarized in the Source data file. Box plots representing the interquartile range (IQR). The top and bottom of the box represent the 75 and 25th percentiles of the data, respectively, with a central line representing the median. Whiskers extend to the upper and lower fences: Upper fence = Q3 + (1.5 × IQR), Lower fence = Q1 − (1.5 × IQR). Outliers are depicted as individual diamond-shaped points beyond the upper and lower fences. **b** Histograms depicting Demeter2 scores per # of cell lines for CRBN (gray) and DCAF1 (blue). **c** Demeter2 scores for selected E3 ligases. Box size definitions are the same than for (**a**). **d** Structure of DCAF1 binder (**13**) and the control compound (**13-N**). **e** SPR binding data of surface immobilized DCAF1(WDR) and (**13**, blue) and

(**13-N**, orange) as analytes. **f** TR-FRET assay binding data of DCAF1(WDR) and (**13**, blue) and (**13-N**, orange) as analytes. Data represents the average and standard deviation of four replicates. **g** Affinity ($K_D$) determined by SPR and $IC_{50}$ determined by DCAF1 TR-FRET for selected DCAF1 binders. **h** Binding mode of (**13**) (orange) in DCAF1 (blue, pdb ID: 8OO5). The inlet with the cartoon depiction of DCAF1 highlights the location of the binding site in the WD40 domain of DCAF1. Shown in the remaining part is a detailed view of selected amino acids interacting with the compound. Hydrogen bonds are indicated as dashes. **i** Chemical proteomics describing the protein interactome of (**16**) in HEK293T cell lysate: Dot plot depicting competition of proteins from (**16**) beads by preincubation with free (**16**) as determined by quantitative proteomics from two independent replicates 1 and 2. Dotted line depicts cut-off at 50% competition. Source data are provided as a Source Data file.

of the piperazine for proteolysis targeting chimeras (PROTACs) exploration and further compound specificity studies. Interestingly, comparison of the binding modes of recently reported DCAF1 ligand **OICR-8268** with (**13**) and (**15**) showed only partial overlap (Supplementary Fig. 1c). To determine cellular compound binding specificity, we coupled an alkyne-analog of (**13**), compound (**16**) to beads using azide-alkyne click-chemistry (Supplementary Fig. 1e). 293T cell lysate incubation with these (**16**)-beads with and without competition of (**16**), to block specific compound interacting protein complexes from binding, enabled the discovery of the whole core CRL4$^{DCAF1}$ complex including CUL4A/B, DDB1, DDA1, and RBX1 (Fig. 1i and Supplementary Data file 2). Interestingly, neither EED, whose early scaffolds led to the discovery of cpd (**13**), nor any other WDR-containing protein (other than DDB1, a known DCAF1 interactor) showed significant competition (>50%) from binding to functionalized beads, demonstrating specificity of our DCAF1 binding scaffold.

## Generation and in-depth genetic and chemical validation of a prototype DCAF1-BRD9 PROTAC

Since DCAF1 is mainly localized to the nucleoplasm[41–43], we first wanted to test a DCAF1 PROTAC against a nuclear target that has been successfully degraded with a TPD MoA. The non-canonical BAF (ncBAF) complex member BRD9 has been shown to be an attractive therapeutic target for e.g., synovial sarcoma[44,45] and has been shown to be amenable for PROTAC-mediated degradation through CRBN and VHL[46,47]. Here, we synthesized a prototype DCAF1-BRD9 PROTAC (**DBr-1**) by coupling the published BRD9 bromodomain binder BI-9564[48] via a piperidine and aliphatic carbon linker to the piperazine of our DCAF1 binding scaffold (**13**) resulting in **DBr-1** (Fig. 2a). The PROTAC molecule showed comparable affinity towards DCAF1 assessed by TR-FRET and SPR measurements, which was improved in presence of an excess of BRD9-BD(130-250) (Fig. 2b, c, Supplementary Fig. 2a, b). To sensitize the TR-FRET assay for the observed higher affinities, we repeated the assay at higher fluorescent tracer concentrations, confirming the strongly enhanced affinity of (**DBr-1**) in the presence of BRD9 (Supplementary Fig. 2c, d). This improved affinity indicates a positive cooperativity as has been previously reported for other ligase-substrate pairs such as VHL and the second Bromodomain of BRD4[49]. Using SPR, we confirmed ternary complex formation between DCAF1, **DBr-1** and BRD9, observing a typical bell-shaped "hook-effect" when reaching saturating ligand concentrations, consistent with respective bimolecular interactions dominating at high **DBr-1** concentrations[50] (Fig. 2d–f).

At 1000 nM **DBr-1** exhibited about 90% BRD9 protein loss after 6 h incubation of HEK293 cells, and thus sub-μM potency as estimated from a dose-response curve (DC$_{50}$ ~500 nM). No BRD9 protein reduction was observed incubating the individual binders at highest concentration for either DCAF1 WDR site (**13**) or the BRD9 BD targeting

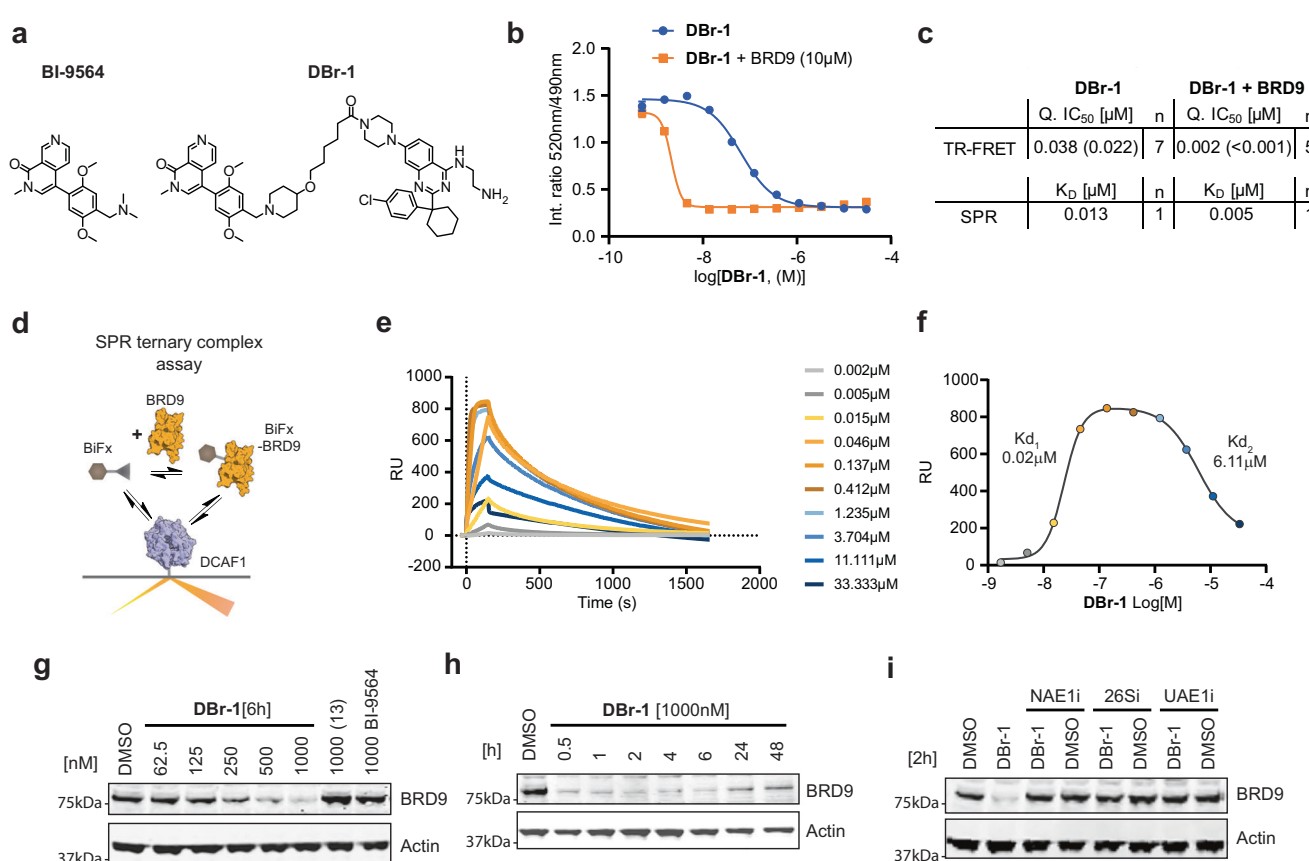

**Fig. 2 | DCAF1-BRD9 PROTAC characterization. a** Compound structures of BRD9 BD binder BI-9564 and corresponding DCAF1-BRD9 PROTAC (**DBr-1**). **b** TR-FRET assay binding data of DCAF1(WDR) and (**DBr-1**, blue) and (**DBr-1** with additional 10 μM BRD9-BD(130-250), orange). **c** Affinity ($K_D$) determined by SPR and IC$_{50}$ determined by DCAF1 TR-FRET for **DBr-1** and **DBr-1** with additional 10 μM BRD9-BD(130-250). **d** Schematic description of SPR ternary binding assay. **e** SPR sensorgrams with surface immobilized DCAF1(WDR) and **DBr-1** as analyte in the presence of 0.2 μM BRD9-BD(130-250). **f** SPR response blotted against **DBr-1** concentration. (Displayed values indicate apparent $K_D$ values of the two separated sigmoidal

transitions, respectively). **g** Immunoblot analysis of HEK293 BRD9-HiBiT/FF/CAS9 cells treated for 6 h with **DBr-1** at various doses as well as (**13**) and BI-9564 at 1000 nM. **h** Immunoblot analysis of HEK293 BRD9-HiBit/FF/CAS9 cells treated with 1000 nM **DBr-1** for various time points. **i** Immunoblot analysis of HEK293 BRD9-HiBiT/FF/CAS9 cells pretreated with NEDD8 E1 inhibitor (NAE1i) [1000 nM], proteasome inhibitor Bortezomib (26Si) [1000 nM] and Ubiquitin E1 inhibitor (UAE1i) [1000 nM] for 2 h, followed by **DBr-1** treatment [1000 nM] for additional 2 h. Source data are provided as a Source Data file.

**BI-9564** (Fig. 2g). A time course with a fixed concentration of 1000 nM **DBr-1** showed rapid degradation (90% degradation already after 30 min) that was maintained for more than 6 h with slow rebound after 24 h and 48 h (Fig. 2h). We observed full rescue of **DBr-1** mediated degradation using either a chemical inhibitor against the ubiquitin-activating E1 (**UAE1i**, TAK-243), the Nedd8-activating E1 (**NAE1i**, MLN4924) or a proteasome inhibitor (**26Si**, bortezomib) (Fig. 2i), demonstrating specificity for the ubiquitin-mediated CRL proteasome pathway.

VHL-based BRD9 PROTAC **VZ185** has been shown to also degrade BRD7 while the CRBN PROTAC **dBRD9** selectively degraded BRD9[46] (Fig. 3a). To assess the selectivity of the DCAF1 PROTAC and benchmark its performance, we tested **DBr-1** in direct comparison with **VZ185** and **dBRD9** for BRD9 and BRD7 degradation in HEK293 cells (Fig. 3b). **DBr-1** showed a similar degradation preference as the CRBN-based PROTAC **dBRD9**, only strongly affecting BRD9 but unlike the **VZ185** not BRD7 levels. We confirmed these findings in HEK293/BRD9-HiBiT and HEK293/BRD7-HiBiT cells, in which **DBr-1** potently degraded BRD9 with a DC50 of 90 nM but only weakly affected BRD7 protein levels (Fig. 3c). Compared to **VZ185** and **dBRD9** the overall degradation efficacy of **DBr-1** for BRD9 was only slightly less potent with a DC50 value of 90 nM (32 nM for **VZ185** and 24 nM for **dBRD9**). To exclude cytotoxicity as a confounding factor for the HiBiT assays, a Cell Titer Glow assay (Promega) was simultaneously performed (Supplementary Fig. 2e). These data highlight the potential for DCAF1 as an alternative ligase with comparable degradation efficacy to the currently almost exclusively used ligases CRBN and VHL.

Next, we synthesized a control degrader using the same methylation strategy described for the control DCAF1 binder (**13-N**) (Supplementary Fig. 3a). Interestingly, the resulting PROTAC **DBr-1-N** showed only a relative reduction of 41× or 27× fold in affinity compared to **DBr-1** as measured by TR-FRET and SPR, respectively (Supplementary Fig. 3b–e). A similar degree of positive cooperativity was also observed for this compound in the presence of an excess of BRD9. When we tested BRD9 levels of HEK293 cells treated with **DBr-1-N**, we observed weaker degradation in both western blots and HiBiT assays which correlated with the loss of DCAF1 binding affinity (Supplementary Fig. 3f–h).

We further wanted to investigate if co-treatment of DCAF1 or BRD9 ligands could rescue **DBr-1** mediated BRD9 degradation. We therefore co-treated BRD9-HiBiT HEK293 cells with **DBr-1** and fixed concentrations of the DCAF1 ligands (**13**) (1 μM), **OICR8268** (10 μM) or the BRD9 ligand **BI-9564** (10 μM). Interestingly, both DCAF1 ligands alone did not affect the degradation efficacy of **DBr-1** (Supplementary Fig. 4a, b). Co-treatment of **BI-9564** was, however able to fully rescue the BRD9 degradation. We speculated that the high level of cooperativity observed for **DBr-1** would potentially require higher levels of DCAF1 binder to outcompete the PROTAC molecule in cells. This notion was supported by the complete rescue of **DBr-1-N** mediated BRD9 degradation with the DCAF1 binders as well as the BRD9 ligand.

To further confirm the DCAF1 mediated mode-of-action and study the underlying ubiquitin pathway components that are hijacked by **DBr-1**, we engineered Hek293/BRD9-HibiT cell line with Cas9 and Firefly luciferase expressing the (BRD9-HiBiT/CAS9/FF), with the aim to enable sgRNA mediated knock-out studies and its impact on BRD9-

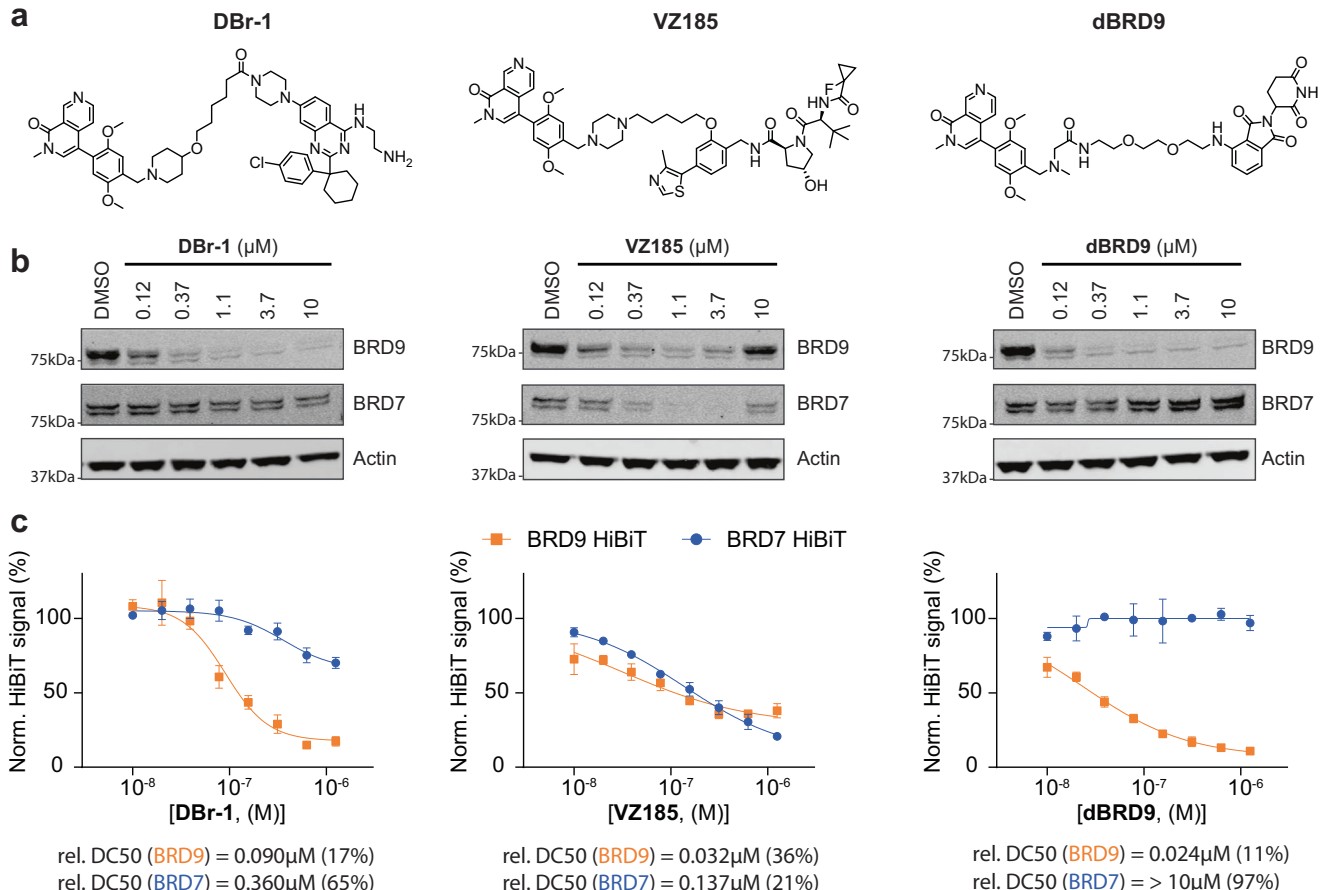

**Fig. 3 | Comparison of different BRD9 PROTACs. a** Compound structures of the DCAF1-BRD9 PROTAC **DBr-1**, the VHL-BRD9 PROTAC **VZ185** and the CRBN-BRD9 PROTAC **dBRD9**. **b** Immunoblot analysis of HEK293 BRD9-HiBiT/FF/CAS9 cells treated for 2 h with **DBr-1**, **VZ185**, and **dBRD9** at various doses. **c** BRD9-HiBiT and BRD7-HiBiT signal detection of samples treated for 2 h with **DBr-1**, **VZ185**, and **dBRD9** at various doses, DC50 values and maximal observed degradation are shown below. Data represents mean ± standard deviation from n = 3 replicates. Source data are provided as a Source Data file.

HiBit levels post **DBr-1** treatment. Incubating these BRD9-HiBit/CAS9/FF cells for 2 h with **DBr-1** confirmed efficient degradation with a $DC_{50}$ of 193 nM and was rescued by UPS inhibitors as previously shown (Fig. 4a). Using three combinations of two individual small guide RNA (sgRNA) against *DCAF1*, one ctrl sgRNA against *BRD9*, one sgRNA against a proteasome pathway ctrl gene *PSMA1* as well as a nonspecific sgRNA against *AAV1*, we genetically confirmed full rescue with all 3 *DCAF1* sgRNA mixes and as such dependence on DCAF1 as the E3 ligase receptor, as predicted (Fig. 4b).

To obtain a more complete picture of ubiquitin pathway components involved in **DBr-1** mediated degradation of BRD9, we performed a CRISPR knock-out rescue screen with an arrayed ubiquitin-sublibrary (943 target genes, 3-4 pooled sgRNAs per gene, Thermo Fisher, Supplementary Data file 3). The **DBr-1** rescue screen was performed as outlined in Fig. 4c. In summary, genetic BRD9-HiBit degradation rescue

was measured after a short 2 h pulse of **DBr-1** in HEK293 BRD9-HiBiT/CAS9/FF cells, 6 days after gene editing by sgRNAs. For hit calling, we set a threshold based on a minimal Firefly signal > log10(1.25). This threshold was based on signal measurements of control wells containing non-infected cells selected with puromycin. The Firefly signal served as control for well-based variation of cell number and was used to normalize the BRD9-HiBiT signals on a plate basis using a non-linear regression 3rd order polynominal curve (Supplementary Fig. 5a – d). From this normalized data, we calculated a robust z-score per gene (Supplementary Fig. 5e) to feed into a gene-level RSA statistical mode[51] to calculate gene effect scores as max. rel. change and significance *P* values (Supplementary Data file 4). We confirmed the strongest rescue with sgRNAs against *DCAF1*, in addition we also found *RBX1*, the RING-box protein 1, a key component of the CRL4DCAF1 complex, as strong hit (Fig. 4d). DDB1 and CUL4A and B, two other key components of the

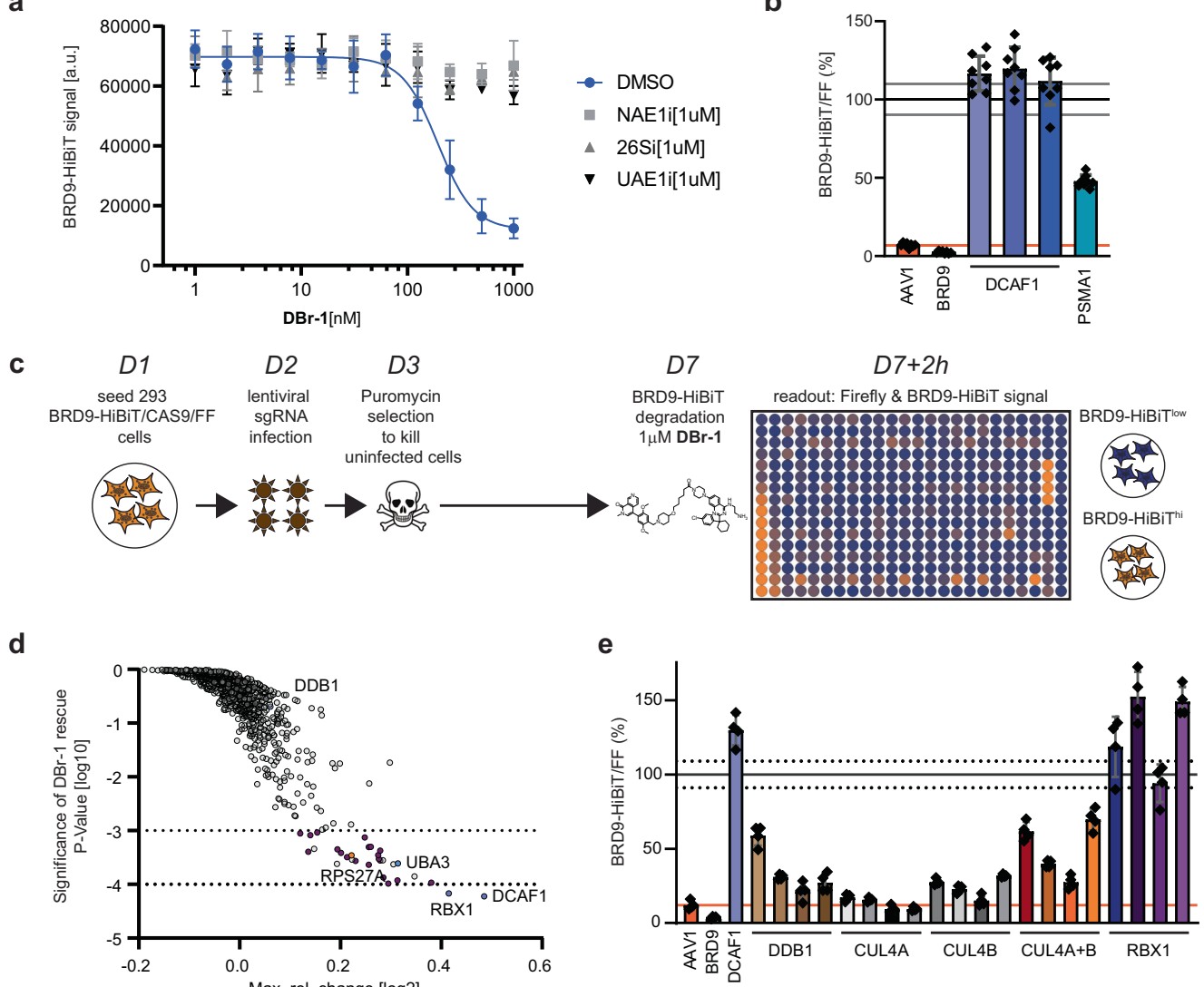

**Fig. 4 | Genetic characterization of the DCAF1-BRD9 PROTAC DBr-1. a** BRD9-HiBiT signal detection of samples pretreated with 1 μM of either NAE1i, 26Si, or UAE1i, followed by 2 h **DBr-1** treatment at various doses, DC50 = 193 nM. Data represents mean ± SEM from *n* = 4 replicates. **b** Relative BRD9-HiBiT vs Firefly signal ratio normalized to non-treated DMSO ctrl after 2 h of **DBr-1** treatment [1000 nM]. Indicated gene editing with sgRNA has been performed as described 6 days before treatment. Data represents mean ± standard deviation from *n* = 8 replicates. **c** Schematic representation and timeline for Ubiquitin-sublibrary sgRNA rescue screen for DCAF1-BRD9 PROTAC **DBr-1**. **d** Ubiquitin sgRNA sublibrary rescue scores

from **DBr-1** treatment plotted as significance of rescue *P* value (y-axis) vs. max. rel. change. Dotted lines at −4 and −3 log10 *P* value indicate strong and weaker hits with a false-discovery rate of 7%. For detailed description of the statistical procedure see the "Method" section. **e** Relative BRD9-HiBiT vs Firefly signal ratio normalized to non-treated DMSO ctrl after 2 h of **DBr-1** treatment [1000 nM]. Indicated gene editing with individual sgRNAs or a combination of two guides for *CUL4A* and *CUL4B (CUL4A + B)* has been performed as described 6 days before treatment. Data represents mean ± standard deviation from *n* = 4 replicates. Source data are provided as a Source Data file.

CRL4[DCAF1] E3 ligase did not rescue from **DBr-1** degradation in our screen setting. CUL4A and B did not pass the Firefly signal threshold together with 117 other genes (including PSMA1 we previously used as ctrl, Supplementary Fig. 5e) and DDB1 scores just above it. In addition, there could be also technical reasons such as low virus titer or limited sgRNA potency in the pool of 4 sgRNAs (Supplementary Data file 3, "Method" section). For further hit-calling, we defined a threshold $P$ value of <log10(−3), based on a randomized activity run (Supplementary Fig. 5f, Method section). Based on this threshold, we discovered 30 genes that rescued **DBr-1** mediated degradation with a false discovery rate (FDR) of 7% (Supplementary Data file 5). In contrast, all the genes that did not pass the Firefly signal threshold did not pass the $P$ value threshold of <log10(−3) as opposed to all the plate-based *DCAF1* ctrl sgRNAs (Supplementary Fig. 5e). Interestingly, *UBA3* and *NAE1*, as well as 20 proteasomal subunit gene knockouts rescue **DBr-1** mediated degradation as previously confirmed with chemical inhibitors NAE1i and 26Si (Supplementary Data file 5 and Fig. 4d). Another interesting hit was *RPS27A*, the ubiquitin and ribosomal protein S27A gene, that expresses a fusion protein between RPS27A and ubiquitin. *RPS27A* is one of the cellular sources of ubiquitin expression, together with *UBB* ($P$ Val log10(−1.2)), *UBC* (P-Val log10(−0.6)) and *UBA52* ($P$ Val log10(−2.1))[52,53]. Taken together, the arrayed genetic rescue screen highly suggested the mode-of-action of **DBr-1** via the CRL4(DCAF1) pathway leading to proteasomal degradation.

To validate RBX1 and follow up on the other CRL4[DCAF1] core components CUL4A, CUL4B, and DDB1 that did not score in the rescue screen, we designed four sgRNAs per gene to test for rescue individually or in the case of the redundant CUL4A and CUL4B[54] also in combination. We could confirm full rescue from **DBr-1** degradation with all four individual sgRNAs against *RBX1*; for *DDB1*, we saw only partial rescue with 1 guide >50%. Interestingly, only combination of *CUL4A* and *CUL4B* targeting guides led to rescue >50%, confirming the redundant nature of these two Cullins (Fig. 4e). In summary, with these chemical and in-depth genetic rescue experiments, we identified CRL4[DCAF1] and proteasome-mediated degradation with our sub-μM DCAF1-BRD9 PROTAC **DBr-1**. These results confirmed that the E3 ligase receptor DCAF1 can be specifically targeted to mediate degradation of the nuclear protein BRD9.

## DCAF1 PROTACs mediate degradation of tyrosine kinases

After successfully degrading BRD9, a nuclear BD-containing protein, we explored whether DCAF1 PROTACs could also mediate degradation of proteins outside the bromodomain family and also outside of the nucleus, since DCAF1 is primarily nuclear. Therefore, we synthesized another prototype PROTAC targeting the tyrosine-kinase subfamily by coupling dasatinib[55] to our DCAF1 binder using a similar piperidine aliphatic carbon linker as for BRD9 (Fig. 5a). Dasatinib binds to >30 kinases[56] and has been shown to inhibit[57] multiple tyrosine kinases of the ABL, DDR, TEC (tyrosine kinase expressed in hepatocellular carcinoma), SRC (sarcoma kinase) and EPH subfamilies at low nM concentrations. Interestingly, these kinase targets include kinases that preferentially localize to the plasma membrane (tyrosine receptor

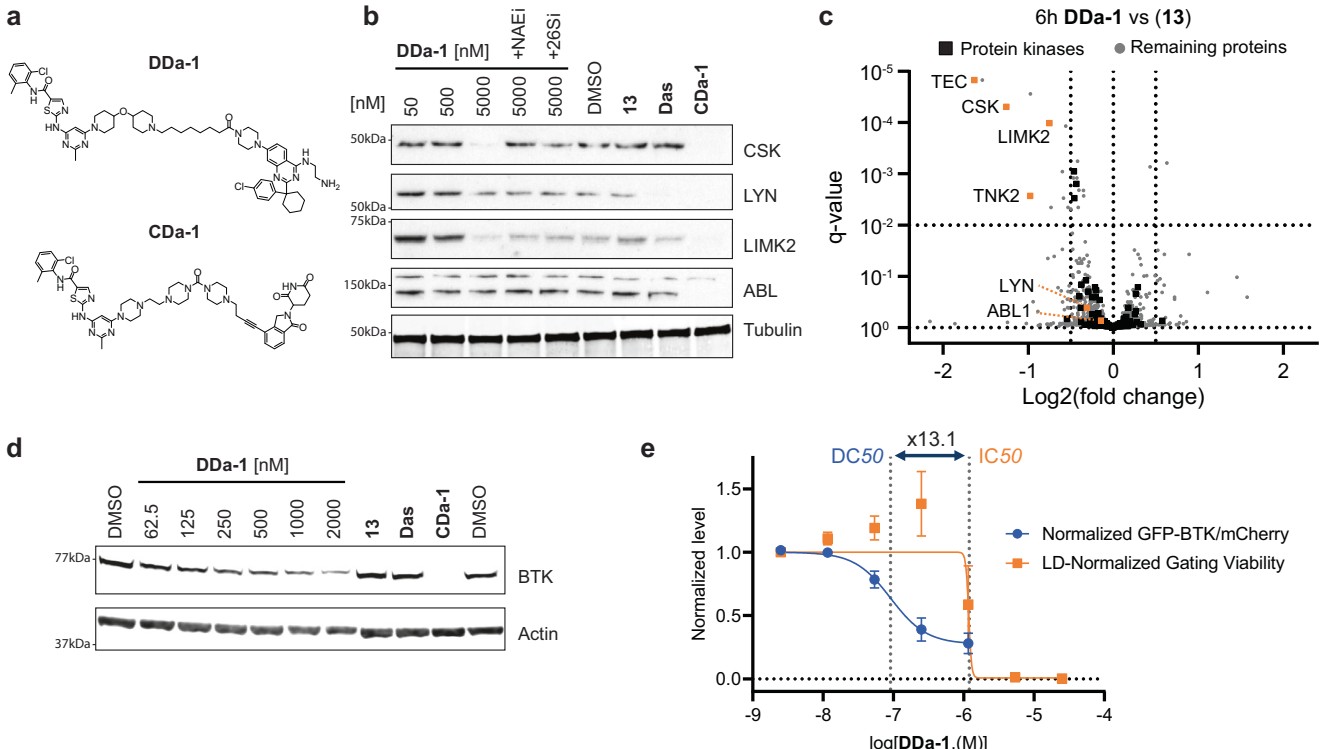

**Fig. 5 | DCAF1-Dasatinib PROTAC degrades multiple tyrosine kinases.**
**a** Compound structure of the DCAF1-Dasatinib PROTAC **DDa-1** and a CRBN-Dasatinib ctrl PROTAC **CDa-1**. **b** Immunoblot analysis of HEK293T cells treated for 6 h with **DDa-1** at 50, 500, and 5000 nM as well as (**13**) and Dasatinib (**Das**) at a dose of 5000 nM as control. Indicated co-treatments with NEDD8 E1 inhibitor (**NAE1i**) [1000 nM] and proteasome inhibitor Bortezomib (**26Si**) [1000 nM] are indicated. CRBN-Dasatinib degrader **CDa-1** [50 nM] served as an internal ctrl degrader. **c** Whole proteomics profile comparing differential protein levels between 5 μM DCAF1-Dasatinib PROTAC **DDa-1** vs 5 μM DCAF1 binder (**13**) after 6 h in HEK293T cells ($n = 3$) plotted as log2 fold change (L2FC, x-axis) versus adjusted $p$ value ($q$ value, $y$ axis). Horizontal dotted line indicates $q$ value $10^{-2}$, vertical lines

indicate a log2 fold change of 0.5. **d** Immunoblot analysis of TMD8 cells treated for 24 h with **DDa-1** at various doses as well as (**13**) and Dasatinib (**Das**) at a dose of 2000 nM as control. CRBN-Dasatinib degrader **CDa-1** [50 nM] served as an internal ctrl degrader. **e** Degradation of BTK-GFP in TMD8 BTK-GFP/mCh cells after 24 h **DDa-1** treatment displayed as normalized rel. change of the ratio between BTK-GFP and mCherry (mCh) signals (blue curve), $DC_{50} = 0.09$ μM. Viability after 24 h **DDa-1** treatment is displayed as rel. change in cellular distribution between viable and apoptotic FSC/SSC gate, $GI50 = 1.20$ μM. The viability window as the ratio between $GI_{50}/DC_{50} = 13.1$ is indicated with dotted vertical lines. Data shown in the graph represents mean ± standard deviation from $n = 3$ replicates. Source data are provided as a Source Data file.

kinases EPH and non-receptor tyrosine kinases TEC and LYN), nucleoplasm (ABL1), cytosol (CSK) or cytosol and nucleoplasm (e.g., LIMK2)[41,58]. This allowed us to study whether DCAF1 demonstrates preference toward cellular localization of its targets. We first tested our DCAF1-Dasatinib PROTAC by Western blotting against four non-receptor tyrosine kinase candidates; C-Src Tyrosine Kinase (CSK), Lck/Yes-Related Protein Tyrosine Kinase (LYN), LIM domain kinase 2 (LIMK2) and Abelson Protein Tyrosine kinase 1 (ABL1). Interestingly, we observed significant reduction in protein levels for CSK and LIMK2, but not for LYN or ABL1 after 6 h in a proteasome (26Si) and CRL-dependent manner (NAE1i) - neither the individual DCAF1 binder nor dasatinib impacted protein levels (Fig. 5b). Compared to a prototype CRBN-Dasatinib degrader, **CDa-1**, which demonstrates degradation of all the four probed kinases, the DCAF1 PROTAC **DDa-1** demonstrates reduced potencies towards the two degraded kinases.

To obtain insights into global protein abundance changes, we analyzed HEK293T cells after 6 h of treatment with **DDa-1** by LC/MS-based proteomics. We were able to confirm downregulation of the two previously studied kinases CSK and LIMK2 and further discovered two additional non-receptor tyrosine kinases, TEC and TNK2, which showed significant protein level reduction (Fig. 5c and Supplementary Data file 6). Interestingly, both TEC and TNK2 are reported to preferentially localize to the plasma membrane[41,59,60].

Bruton's tyrosine kinase (BTK), another TEC family member, has been successfully targeted in a variety of blood cancers with the covalent BTK inhibitor ibrutinib in the clinic[61–63]. Furthermore, BTK has been shown to be an attractive target for PROTAC mediated degradation hijacking the ligase CRBN[6,64] with advanced TPD molecules in clinical trials (such as e.g., NX-2127[65]). Since BTK is not expressed in HEK293 cells but has been shown to be bound and inhibited by dasatinib[56,66] we wanted to test if our dasatinib-DCAF1 PROTAC was active against this kinase. To this aim, we probed the BTK-expressing diffuse large B-cell lymphoma (DLBCL) cell line TMD8[67] with **DDa-1** at various concentrations and could confirm BTK degradation with sub-$\mu$M potency (Fig. 5d). To further explore BTK degradation in a more quantitative, higher-throughput profiling system, we engineered TMD8 cells stably expressing a bicistronic BTK-GFP/mCherry construct (referred to as TMD8 BTK-GFP/mCh). Ratiometric analysis of GFP to mCherry signal changes allows a quantitative measurement of degradation, uncoupling degradation from transcriptional and translational effects[68,69]. TMD8 DLBCL cells have been shown to be dependent on BTK and hence sensitive to BTK inhibition[70] and degradation[71].

To ensure that our engineered TMD8 BTK-GFP/mCh cells still respond to BTK perturbations, we used a BTK inhibitor BTKi (17) (WO2019/186343 A1) and compared CYLD tumor suppressor accumulation[72]. Reassuringly, parental TMD8 and engineered TMD8 BTK-GFP/mCh cells gave a very similar biological response to BTK inhibition after 24 h (Supplementary Fig. 6a).

Considering the sensitivity of TMD8 BTK-GFP/mCh cells to BTK perturbations, we explored potential signal changes impacting cellular fitness to ensure confidence in measuring BTK degradation. We observed that the TMD8 BTK-GFP/mCh cells show some response to on-target BTK inhibition when measuring forward and sideward scattering by flow cytometry (FSC and SSC) of single cells after BTKi (17) treatment within the 24 h timeframe that we used to measure degradation effects. At very high BTKi (17) concentrations [25 $\mu$M], we observed significant population shifts in FSC and SSC. By using two apoptotic markers for early and late apoptosis, we could confirm that the apoptotic cells almost exclusively localize to the FSC/SSC-shifted population (Supplementary Fig. 6b). Therefore, our gating strategy to score for degradation focused on the viable cell population as defined by the FSC/SSC-scatter population observed in DMSO-treated cells. In addition, we set a minimal viability threshold of 25% viable cells and excluded treatments from degradation calculation due to the low number of viable cells observed (Supplementary Data file 7, "Max dose

w/ Gating Viability >25%"). Having established this additional cell viability read-out when measuring BTK-GFP degradation allowed us to calculate a parameter to confidently score degradation. Comparing the change of BTK-GFP/mCherry signals (DC$_{50}$) and viability (GI$_{50}$) allowed calculation of the fold change of viability GI$_{50}$ versus degradation DC$_{50}$ (Supplementary Data file 7, "V/D [GI$_{50}$ Viability/DC$_{50}$ Degradation]"). This metric simplified distinguishing on-target (as seen for BTKi (17)), as well as potential off-target effects on cellular fitness that could impair the degradation read-out. BTKi (17) has a V/D score of 0.3 whereas the very potent degrader **CDa-1** has a score of 2124.8; when testing our DCAF1 binder (13), we observed loss of viability at concentrations of >5 $\mu$M with a V/D score of 0.2 (Supplementary Data file 7, Supplementary Fig. 3c, d). Finally, testing **DDa-1** in this system validated our findings on endogenous BTK degradation (Fig. 5e, Supplementary Data file 7: DC$_{50}$ = 0.09 $\mu$M and V/D = 13.1).

## Discovery of a potent BTK-specific DCAF1 PROTAC

Next, we investigated the potential to develop more specific DCAF1-directed BTK PROTACs. To this end, we synthesized different DCAF1 PROTACs with a specific BTK binder (17) (WO2021/55295) and a close analog. From a small library of synthesized PROTACs, two molecules were selected for further characterization based on their ability to potently degrade BTK in the TMD8 BTK-GFP/mCh cellular assay - **DBt-5** incorporated a flexible PEG-based linker while **DBt-10** bridged the DCAF1 and BTK binding moieties by a more riged scaffold (Fig. 6a).

We compared those two PROTAC molecules by characterizing them in a variety of assays. Binary affinities for DCAF1 were assessed by TR-FRET and SPR, in presence or absence of additional BTK (Fig. 6b, Supplementary Fig. 7a, c and Supplementary Fig. 8). **DBt-5** showed a slightly higher affinity than **DBt-10** for DCAF1 in both assays. However, additional BTK in the TR-FRET and SPR assay increased the affinity for DCAF1 only for **DBt-10**, while decreasing it for the more flexible **DBt-5**. This lack of positive cooperativity of **DBt-5** was further reflected in the shorter ternary complex half-life compared to **DBt-10** (Supplementary Fig. 8). Binary affinities towards BTK were tested in a biochemical assay (Supplementary Fig. 9a, Supplementary Data file 7) and in cellular BTK NanoBRET target engagement assay. The latter were performed also with additional Digitonin treatment, allowing a relative assessment of membrane permeability of the tested compounds[73]. While **DBt-5** showed potent BTK engagement in both intact and permeabilized cells, **DBt-10** showed reduced BTK target engagement without prior membrane permeabilization assay (Supplementary Fig. 9b, Supplementary Data file 7). This indicated a less optimal permeability of **DBt-10** compared to the larger and more flexible **DBt-5**.

We further assessed the ternary complex formation of DCAF1-PROTAC-BTK and the in-vitro ubiquitination of BTK by Cul4-DCAF1 upon treatment with the two PROTACs. Interestingly, higher concentrations of **DBt-5** compared to **DBt-10** were required to reach the maximal SPR response. The maximal initial velocity of the ubiquitination reaction overlapped well with the maximum of ternary complex for **DBt-5** (Fig. 6c) while this maximal initial velocity occurred for **DBt-10** at higher concentrations in respect to the complex formation. Interestingly, the initial velocity and the maximal SPR response were higher for the more rigid **DBt-10**, suggesting that the more stable ternary complex could enhance the ubiquitination rate (Supplementary Fig. 7b, c).

Despite displaying different binary and ternary complex stabilities and cell permeabilities, both **DBt-5** and **DBt-10** potently degraded BTK-GFP in the aforementioned TMD8 BTK-GFP/mCherry cell-based assay (Fig. 6b, d, Supplementary Data file 7). High V/D [GI$_{50}$ Viability/DC$_{50}$ Degradation] values were measured for both PROTACs, with **DBt-10** exhibiting less toxicity than **DBt-5**. Therefore, we selected **DBt-10** for further characterization.

We confirmed the high viability score of **DBt-10** by measuring early and late apoptotic cells (Supplementary Fig. 10a), and

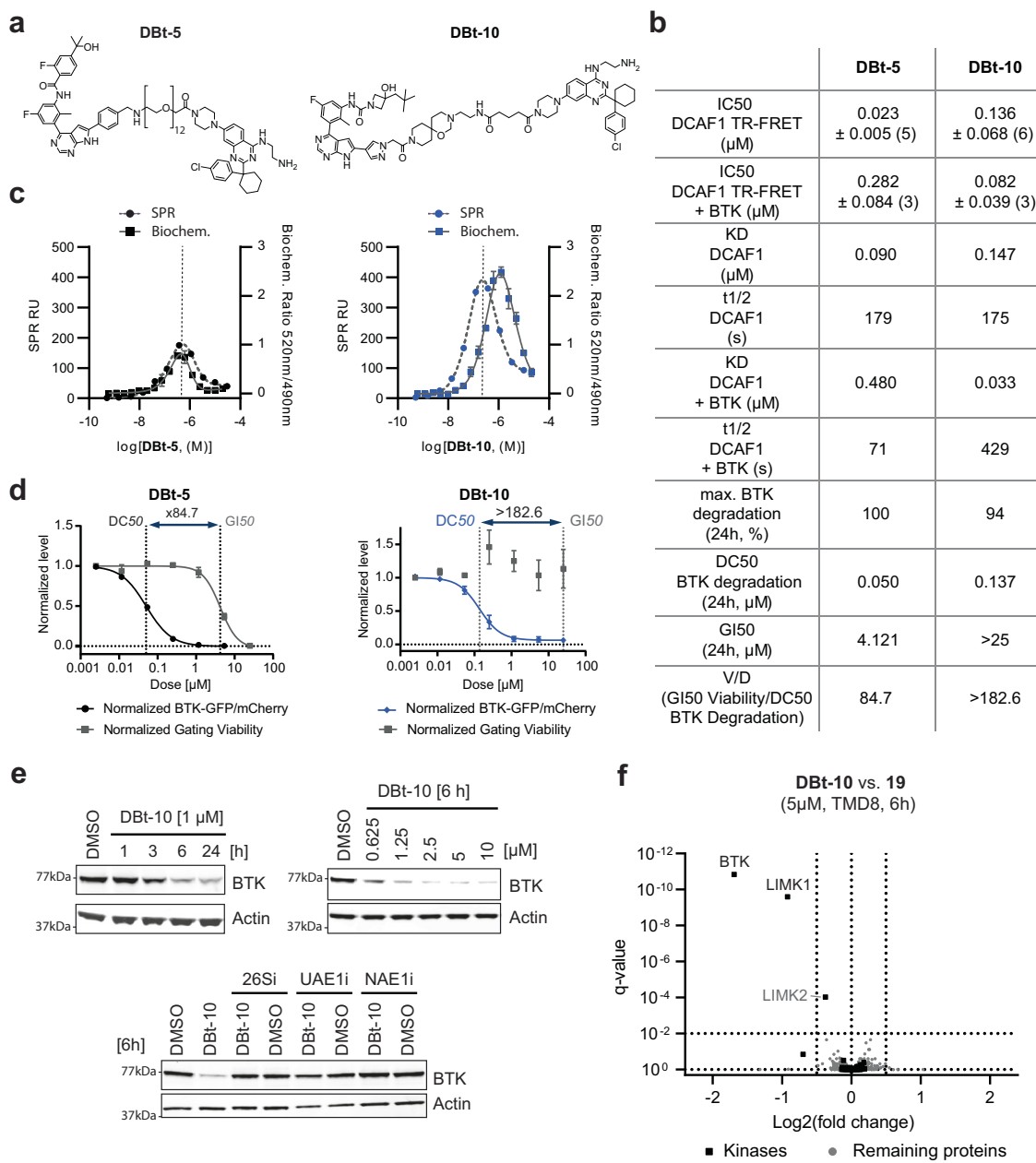

**Fig. 6 | Characterization and discovery of DCAF1-BTK PROTACs. a** Chemical structures of the DCAF1-BTK PROTACS **DBt5** and **DBt-10**. **b** Characterization summary of various assays profiling the DCAF1-BTK PROTACs **DBt5** and **DBt-10**. **c** Overlay of SPR ternary complex formation assays (dashed lines, left y-axis) with biochemical ubiquitination rates of DCAF1 measured in ubiquitin-transfer based TR-FRET assay (right y-axis, mean ± standard deviation from $n = 3$ replicates). **d** BTK degradation and cellular viability (dashed lines) assessed in BTK-GFP/mCh TMD8 cells after 24 h. The data points have been normalized to DMSO controls and each point represents the average and standard deviation of three independent experiments in triplicates. The viability window as the ratio between $GI_{50}/DC_{50}$ is indicated with dotted vertical lines. **e** Immunoblot analysis of TMD8 cells treated with 1 μM **DBt-10** for 1, 3, 6, and 24 h time points, treated with various doses of **D Bt-10** for 6 h and TMD8 cells pretreated with proteasome inhibitor Bortezomib (**26Si**) [1000 nM], Ubiquitin E1 inhibitor (**UAE1i**) [1000 nM] and NEDD8 E1 inhibitor (**NAE1i**) [1000 nM] for 30 min, followed by **DBt-10** treatment [2500 nM] for additional 6 h. **f** Proteomic analysis of TMD8 cells treated ($n = 3$) for 6 h either with 5 μM **DBt-10** or 5 μM (**19**). Highlighted are proteins with a log2 fold change <−0.5 and a $q$ value < 0.01. Detected kinases are represented as squares, while all other proteins are shown as dots. Source data are provided as a Source Data file.

after 24 h we did not observe a strong induction of apoptosis as compared to BTKi (**17**) (Supplementary Fig. 6b), potentially due to other different off-targets of the degrader or the underlying different cell permeability and kinetic properties of inhibition versus degradation. We observed efficient degradation of endogenous BTK in TMD8 cells at around 6 h (Fig. 6e) whereas control (**13**) treatment did not reduce BTK at any time points (Supplementary Fig. 5b). A dose-response at the 6 h time point showed that we drive significant reduction of BTK at a concentration of

2.5 μM (Fig. 6e), which could be chemically rescued using 26Si, NAE1i and UAE1i to validate proteasome, CRL and ubiquitination dependency of **DBt-10** mediated BTK degradation, respectively (Fig. 6e). We could further confirm the DCAF1 dependency by co-treating TMD8 BTK-GFP/mCherry with both **DBt-10** and various DCAF1 and BTK ligands. The DCAF1 ligands (**13**) and (**16**) rescued **DBt-10** mediated BTK degradation to a similar extent as the BTK ligands (**17**) and (**18**) (Supplementary Fig. 9e–d). The DCAF1-BRD9 PROTAC **DBr-1**, the alternative DCAF1 ligand **OICR-8268**, the

PEG-linked DCAF1 binder (**15**) and the BTK-linker fusion (**19**) also impaired BTK degradation, but to a lesser extent than (**13**), (**16**) and (**17**). The control compounds (**13-N**) and **DBr-1-N** exhibited no measurable effect on DBt-10-mediated BTK degradation in this assay.

To assess the selective degradation of BTK, we performed a proteomic analysis of TMD8 cells treated for 6 h with either 5 μM **DBt-10** or (**19**). We confirmed BTK as the strongest reduced protein among those proteins detected after **DBt-10** treatment (Fig. 6f, Supplementary Data file 8). LIM domain kinase 1 (LIMK1) was the only other protein with significantly reduced abundance in the **DBt-10**/TMD8 sample, underlining that DCAF1-ligands can also be used to develop highly selective PROTACs.

### DCAF1-PROTACs provide an alternative for cells lacking VHL expression or with acquired resistance to CRBN-based degraders

Next, we investigated if the developed DCAF1-PROTACs could be used in cases in which VHL or CRBN-based PROTACs lack efficacy due to intrinsic or acquired resistances (Fig. 7a). Therefore, we chose 768-O cells intrinsically lacking VHL-expression[74] as a model system to test if DCAF1-PROTACs could be used in such case as an alternative to degraders based on VHL. Assessing BRD9 levels in 768-O cells treated with **VZ185** confirmed the intrinsic resistance against VHL-PROTACs in this cell line (Fig. 7b). The DCAF1 and CRBN-based BRD9 degraders **DBr-1** and **dBRD9** exhibited comparable efficient degradation of BRD9 in the same experimental setting (Fig. 7b). These results validate DCAF1 as an alternative essential ligase for the application of PROTACs in biological contexts where VHL is not expressed, e.g., in renal cell carcinoma.

In addition to intrinsic resistances due the lack of expression of the E3, there is emerging evidence of acquired resistance to TPD drugs due to mutations of both the ubiquitination machinery including the E3 ligase as well as the targeted protein (Fig. 7a). Such adaptive mechanisms have also been found in patients treated with IMiDs leading to reduced clinical response due to changes in CRBN-mediated degradation of the targeted POIs[20,75]. Anticipating that such resistant mechanism will not be limited to molecular glues, we wanted to test if

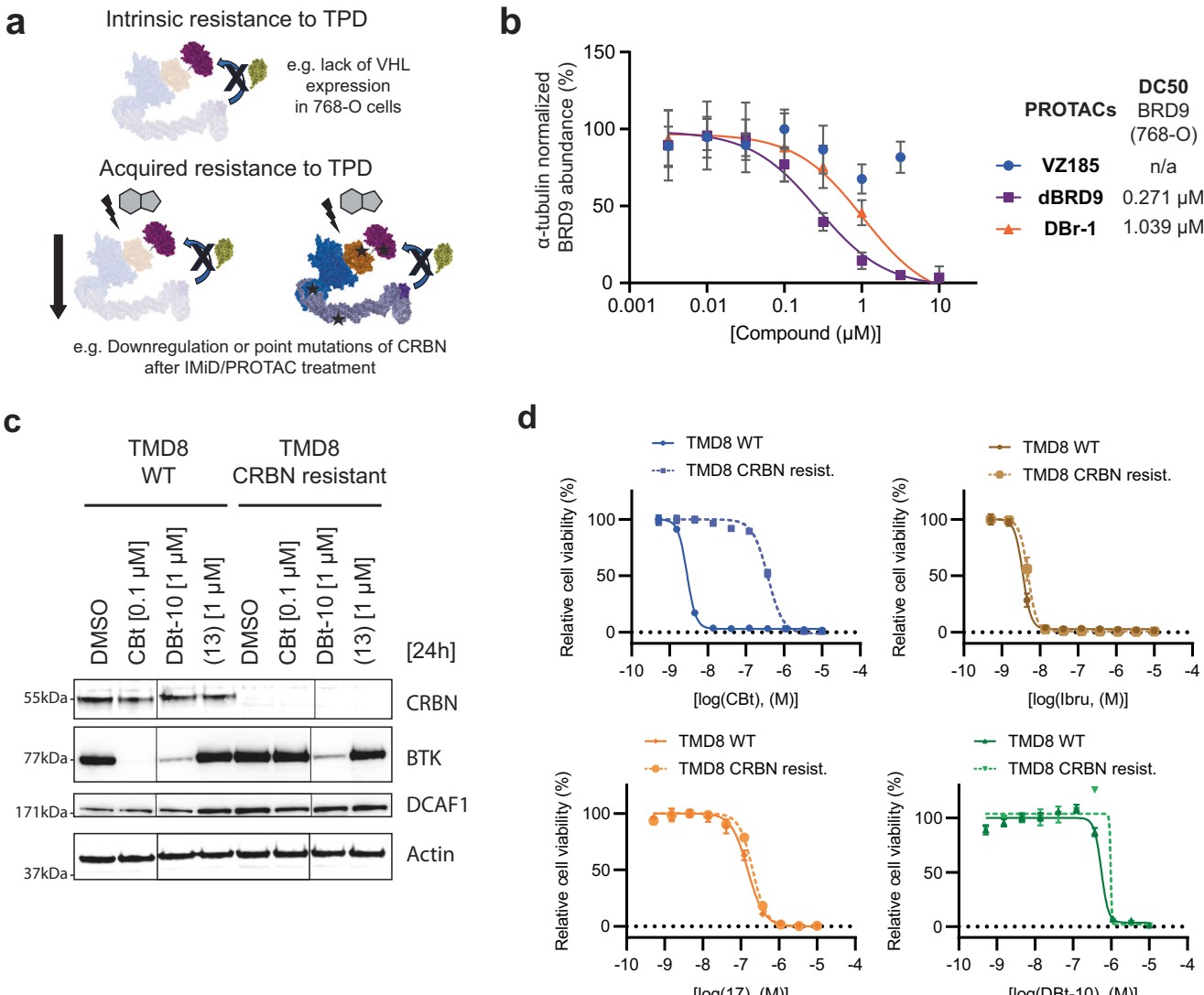

**Fig. 7 | Characterization of DCAF1-based PROTACs in cells resistant to VHL or CRBN-mediated targeted protein degradation. a** Schematic illustration of examples of intrinsic and acquired resistance mechanisms to targeted protein degradation (TPD). **b** 768-O cells treated for 2 h with **DBr-1**, **VZ185**, and **dBRD9** at various doses. Shown is the change of relative BRD9 abundance normalized to the DMSO control. Data represents average ± standard deviation of *n* = 3 replicates and calculated DC50 values of BRD9 are displayed. **c** Immunoblot analysis of TMD8 WT and TMD8 CRBN resistant cells after 0.1 μM **CBt** and 1 μM **DBt-10** treatment for 24 h. **d** Viability measured by CellTiter-Glo for **CBt**, Ibrutinib (**Ibru**), BTKi (**17**) and **DBt-10** in WT (solid) vs CRBN resist. (dotted) TMD8 cells. Data represents mean ± standard deviation from *n* = 3 replicates. Source data are provided as a Source Data file.

the essential ligase DCAF1 could serve as substitute in cases where CRBN-based PROTACs lost their degradation efficacy. This was particularly interesting since both DCAF1 and CRBN require the same binding partners, such as DDB1 and CUL4A, to form the active E3 ligase complex. Therefore, in order to study **DBt-10** in a CRBN-BTK PROTAC resistant setting, we generated TMD8 cells resistant to CRBN-BTK mediated PROTACs by prolonged treatment with escalating suboptimal doses of a recently patented CRBN-BTK PROTAC **CBt** (WO2021/53495 A1). Probing both TMD8 cell settings (**CBt** sensitive vs resistant) with **CBt** enabled BTK degradation only in the WT but not in the **CBt**-resistant TMD8 cells, in line with loss of CRBN expression in the latter (Supplementary Fig. 10c).

Probing these TMD8 **CBt** sensitive and resistant cells with the DCAF1-BTK PROTAC **DBt-10** enabled equipotent degradation of BTK, consistent with retention of DCAF1 protein levels independent of CRBN status (Fig. 7c). Next, we assessed if degradation or lack thereof translates into viability effects in the TMD8 **CBt**-sensitive and -resistant cells when incubating with **CBt** or **DBt-10**. As expected, treatment of these two TMD8 cell lines with **CBt** led to a 100-fold difference in proliferation inhibition, as BTK PROTACs bind to BTK via an inhibitor and BTK inhibition is not impaired by loss of CRBN. We confirmed a CRBN-mediated resistance mechanism by probing both **CBt** sensitive and resistant TMD8 cells with BTK inhibitors ibrutinib (**Ibru**) or (**17**), and neither inhibitor showed any relative potency change between the two different cell settings, thus proving that BTK catalytic function is still intact and is not the cause of resistance. Intriguingly, treatment with **DBt-10** also did not show any relative potency differences in proliferation between **CBt** sensitive and resistant settings (Fig. 7d, Supplementary figures. 10d/e). The DCAF1 binder (**13**) did not show any reduction of BTK as well, as we do observe a fivefold window between **DBt-10** vs (**13**) treatment, which makes us believe that the impact on TMD8 cell proliferation by **DBt-10** is mediated through BTK degradation.

## Discussion

In this study we took a non-covalent DCAF1 WDR binder[25] and converted into PROTAC E3 ligase anchor. We selected DCAF1 as our ligase of choice due to its essentiality within the WDR containing E3 ligase receptor family, with the hypothesis that this might reduce the chance of ligase-specific resistance mechanisms through e.g., loss of ligase expression. DCAF1 as an E3 ligase has been targeted previously by covalent PROTACs[19], although in this study, de novo substrate degradation was only possible when DCAF1 was overexpressed from a strong CMV-promoter but not at endogenous levels of DCAF1. Furthermore, the covalent ligand MY-1B showed only partial cellular engagement of DCAF1. The authors of this publication stated that most likely more optimized DCAF1 ligands would help to degrade targets with lower, endogenous levels of DCAF1. Since the CRL4$^{DCAF1}$ ligase complex has been structurally described to be in a tetrameric, auto-inhibited state, in addition to a dimeric, active conformation favored through substrate binding[35], it could additionally be possible that DCAF1 overexpression also increases the total amount of active CRL4$^{DCAF1}$ ligase complex. Our study shows that PROTACs with a reversible non-covalent double digit nM binder to the DCAF1 WDR donut-hole can hijack endogenous DCAF1 and mediate degradation of target proteins, confirming accessibility of active CRL4$^{DCAF1}$ ligase at physiological expression levels.

We also designed a suitable control compound (**13-N**) with an over 100x fold reduced binding affinity for DCAF1. Additionally, we linked this di-methylated analog with **BI-9564**, creating the PROTAC **DBr-1-N** with weakened degradation efficacy for BRD9. However, the strong loss of binary affinity of (**13-N**) towards DCAF1 was only partially observed for **DBr-1-N** (27x affinity reduction for DBt-1-N compared to the **DBr-1** binding affinity). Interestingly, we further measured a similar high degree of cooperativity for this compound as for **DBr-1**,

suggesting additional DCAF1-BRD9 interactions apart of the ones facilitated by both PROTAC molecules. The elucidation of the exact nature of such protein-protein interactions will require future structural studies beyond the scope of this work. We further confirmed that the remaining, weak BRD9 degradation of **DBr-1-N** was DCAF1 dependent by competition experiments with the DCAF1 binders (**13**) and **OICR-8268**. These results and the rescue of BTK degradation validated the cellular target engagement of DCAF1 ligands as previously reported[31] and highlighted the benefits of high-quality ligands of different chemical scaffolds.

Our degradation studies with BRD9 and tyrosine kinases (dasatinib and BTK) targeting PROTACs showed that although DCAF1 is predominantly localized in the nucleus, it can degrade cytosolic and even membrane localized tyrosine kinases such as CSK or TEC, respectively. The comparison of substrate preference of BRD9 and BRD7 with VHL and CRBN PROTACs revealed a similar preference for BRD9 as observed for the CRBN-based **dBRD9**. However, further comparative studies with a larger number of POIs are needed to better understand how effects such as relative cellular localizations of DCAF1 and the recruited substrates can impact the degradation-efficacy of DCAF1 PROTACs.

After establishing that endogenous DCAF1 can degrade a variety of de novo substrates at sub-μM potencies, we wanted to expand our studies to BTK-specific degraders, a target that has been extensively studied and characterized in hematological cancers[61–63], with the aim to discover potent DCAF1-BTK PROTACs and test if these can overcome acquired resistance to CRBN-BTK PROTACs. One important confounding factor when establishing a BTK degradation assay was that our DCAF1 binder (**13**) as well as some of the DCAF1 PROTACs lead to cellular toxicity at concentrations above 5–10 μM. Therefore, we established a viability score in our flow cytometry-based assays to enable confident degradation data reporting (Supplementary Fig. 3). Our discovery approach led to the potent DCAF1-BTK PROTAC **DBt-10** with minimal off-target toxicities and high selectivity for BTK. Interestingly, **DBt-5**, incorporating a longer, linear linker, displayed worse ternary complex formation and in-vitro ubiquitination with recombinant CRL4$^{DCAF1}$ than **DBt-10** but better, relative cell permeability. However, both PROTACs degraded BTK with a similar potency and extent. Our in-depth DCAF1-BTK degrader characterization further supports many previous observations in the TPD field that efficient PROTAC discovery cannot currently be predicted through simple rules and single parameters. Successful PROTAC discovery is a multi-parametric empiric optimization process, which in our case finally led to the discovery of the DCAF1-BTK PROTAC **DBt-10** with sufficient potency to enable studies in a CRBN-BTK PROTAC-resistant setting.

Indeed, with this optimized PROTAC **DBt-10** we could show that BTK is still degraded with comparable efficiency in cells with acquired resistance to CBRN-BTK PROTACs due to loss of CRBN expression. This translated to equipotent inhibition of cell proliferation independent of CRBN status in these BTK dependent DLBCL cell line. Additionally, we could show that the DCAF1-BRD9 PROTAC **DBr-1** is able to degrade in BRD9 in a cell line intrinsically lacking VHL expression and thus DCAF1 could provide a viable alternative to VHL for PROTAC development. These observations demonstrate that targeting an alternative ligase such as DCAF1 in intrinsic or acquired resistances for PROTAC molecules is a viable option to overcome these resistance mechanisms and might provide a strategy for combinations of degraders which engage different ligases.

## Methods
### Experimental model and subject details
Cells used: HEK293T cells (donor sex: female),HEK293/BRD9-HiBiT KI pool (PROMEGA #CS3023119) and HEK293/BRD7-HiBiT KI pool (PROMEGA # CS3023124), were cultured in DMEM (Gibco cat#11995-065) supplemented with 10% fetal bovine serum (FBS, Avantor cat#97068-085). TMD8[67] (donor sex: male) and TMD8/BTK-GFP/mCh were

cultured in IMDM (Gibco cat# 12440-053) supplemented with 10% fetal bovine serum (FBS). The cell lines were not further authenticated. An additional overview of all models and resources can be found in Supplementary Data file 10.

## Extraction of genetic dependencies of WDR E3 ligases

We manually annotated all E3 ligase related proteins and function from a published resource of 361 WDR containing proteins[30] and further manually classified the E3 subfamilies (Supplementary Data file 1). From this table we extracted Achilles Demeter2 scores from the DepMap 22Q2 release (D2_combined_gene_dep_scores.csv at https://depmap.org/portal/download/all/). Downloads, data handling, and plotting were performed using Python 3.9.5.

## Protein production

His6-TEV-DCAF1(1039-1401)Q1250L, His-TEV-DCAF1(1079-1393) Q1250L, Avi_DCAF1(1073-1399)_E1398S_His, Strep-PreSc-DCAF1(987-1507), DDB1(1-1140), His-TEV-CUL4(38-759), His-TEV-Rbx1(12-108), His_PreSc_BTK (389-659), His_PreSc_Avi_BTK (389-659) were all subcloned into pIEx/Bac-3 derived vectors for expression in Sf9 cells using the flashBAC ™ system. Recombinant baculoviruses were generated and amplified followed by large-scale expression in Sf9 cells as previously described[76]. Transfected cells were typically grown at 27 °C for up to 62 h before harvesting by centrifugation. For in-vivo biotinylation BirA was co-transfected.

Cell pellets were resuspended in lysis buffer (see Table 1) supplemented with o'complete (EDTA free) protease inhibitor cocktail (Roche) and Benzonase and lysed by sonication or use of a microfluidizer. After removal of cell debris by centrifugation at 4 °C at 50,000 × *g* for at least 30 min the cleared supernatants were purified using Ni-NTA columns (Cytivia) on AKTA™ systems (Cytivia). For proteins containing cleavable affinity tags, C3 protease (PreScission™) or TEV protease were used to cleave the tag. Proteins were further purified by Ion exchange chromatography (see Methode Table 1) followed by a final Size Exclusion Chromatography step on SuperdexS200 or SuperdexS75 columns (Cytivita). Proteins were concentrated, aliquoted, and flash-frozen in liquid nitrogen.

The DCAF1//DDB1//CUL4A//RBX1 complex was expressed, purified and neddylated as previously described[35]. The enzymes PPBP1-UBA3 and UBC12 as well as the substrate NEDD8 were expressed and purified as reported before[77–79].

BRD9 (130-250) was subcloned into a pGEX6P1 vector and transformed into E.Coli Rosetta cells. Cells were grown at 37°C to an OD600 of 0.7-1.0 in Terrific Broth (Teknova cat# G8110). Expression was induced with 0.2 mM IPTG at 20°C overnight. Cells were harvested by centrifugation. The cell pellet was resuspended in lysis buffer (see Table 1) and cells were lysed by passing through a microfluidizer. Cell debris was removed by centrifugation at 4°C. Affinity chromatography was performed on Glutathione beads (GE Bioscience cat# 17-0756-05) and successively the affinity tag was cleaved by addition of C3 protease (PreScission™) while dialyzing against 1xPBS pH 7.4, 400 mM KCl, 5 mM DTT. After a reverse affinity step with fresh GST beads the protein was applied to a Superdex S200 column equilibrated in the final buffer (see Table 1). The protein was concentrated, aliquoted, and flash-frozen in liquid nitrogen.

## Surface plasmon resonance (SPR)

Surface plasmon resonance (SPR) experiments were performed on Biacore T200 and 8 K+ instruments (Cytiva). Biotinylated Avi_D-CAF1(1073-1399)E1398S-His6 was immobilized (~1000–4000 RUs) on a SA chip. Compounds were diluted in the SPR running buffer (20 mM HEPES, pH 8, 200 mM NaCl, 0.5 mM TCEP, 0.01% Tween-20, 5% Glycerol) for binary compound-DCAF1 complex assessments (typically in a range from 0.003 μM up to 100 μM). For the measurement of the

**Table 1 | Purification details of recombinant proteins used**

| Protein | Expression system | Lysis buffer | Ion-exchange column | Final buffer |
|---|---|---|---|---|
| His6-TEV-DCAF1(1039-1401)Q1250L | Sf9 insect cells | 50 mM Tris pH 8.0, 200 mM NaCl, 0.1% Triton X-100, 4 mM TCEP | ResourceQ | 20 mM Hepes pH 7.5, 200 mM NaCl, 2 mM TCEP |
| His-TEV-DCAF1(1079-1393)Q1250L | Sf9 insect cells | 50 mM Tris pH 8.0, 200 mM NaCl, 0.1% Triton X-100, 4 mM TCEP | ResourceQ | 20 mM Hepes pH 7.5, 200 mM NaCl, 2 mM TCEP |
| Avi_DCAF1(1073-1399)E1398S-His6 | Sf9 insect cells Coexpressed with BirA | 50 mM Tris (pH 8.0), 300 mM NaCl, 10% Glycerol (w/v), 5 mM TCEP, 5 mM Imidazole | Q HP | 50 mM Tris, 300 mM NaCl, 10% Glycerol, 2 mM TCEP (pH 8.0) |
| DCAF1(1073-1399)E1398S-Avi-His6 | Sf9 insect cells Coexpressed with BirA | 50 mM Tris (pH 8.0), 300 mM NaCl, 10% Glycerol (w/v), 5 mM TCEP, 5 mM Imidazole | Q HP | 50 mM Tris, 300 mM NaCl, 10% Glycerol, 2 mM TCEP (pH 8.0) |
| Strep-PreSc-DCAF1(987-1507)//DDB1(1-1140)//His-TEV-CUL4(38-759)//His-TEV-Rbx1(12-108) | Sf9 insect cells | 50 mM Tris pH 8.0, 200 mM NaCl, 4 mM TCEP | ResourceQ | 50 mM HEPES pH 7.4, 200 mM NaCl, and 0.25 mM TCEP |
| His_PreSc_BTK (389-659) | Sf9 insect cells | 50 mM Tris pH 7.6, 400 mM NaCl, 10% Glycerol, 3 mM TCEP | ResourceQ | 20 mM TrisHCl, 100 mM NaCl, 3 mM TCEP, 1 mM EDTA, pH 8.0 |
| His_PreSc_Avi_Btk (389-659) | Sf9 insect cells Coexpressed with BirA | 50 mM Tris pH 7.6, 400 mM NaCl, 10% Glycerol, 3 mM TCEP | ResourceQ | 50 mM Tris pH 8.0, 200 mM NaCl, 5% Glycerol, 2 mM TCEP |
| GST-PreSc-BRD9 (130-250) | E.Coli | 1x PBS pH 7.4, 400 mM KCl, 5 mM DTT, 1 mM MgCl₂, 10% Glycerol, 1 mM PMSF | n/a | 25 mM HEPES pH 7.5, 100 mM KCl, 5 mM DTT |

VHF543-DCAF1 complex on the Biacore T200 samples were applied to the flow cell at a flow rate of 30 μL/min. The association was assessed for 60 s and the dissociation was monitored for 410 s. For measurements of all other compounds and ternary complexes on the Biacore 8K+ the association was assessed for 150 s and the dissociation was monitored for 1510 s or longer. For the ternary complexes, the compounds were diluted in the SPR running buffer supplemented with 0.2 μM or 5 μM BTK (389-659) or BRD9 (130–250). Binary complexes were analyzed with a kinetic 1:1 fit model to obtain $K_D$ values. The fit of apparent steady state RU intensities of the ternary complexes was performed using a bell-shaped curve algorithm implemented in Graphpad Prism. For the off-rate determination of the ternary complexes only the datapoints of compound concentrations contributing to the transition at lower compound concentration were included in a 1:1 dissociation fit model. Residence times ($t_{1/2}$) were calculated by the inversion of the off-rate.

## X-ray crystallography

The DCAF1 apo crystals were grown by mixing cleaved His-TEV-DCAF1(1079-1393)Q1250L at 9 mg/mL with 1.91 M LiSO4 and 0.1 M Tris pH 7.5 in a 1:1 ratio (200 nL + 200nL) in sitting drop crystallization plates at 20 °C. Crystals were soaked with 1 mM (**13**) for 15 mins and cryo-protected by the addition of 20% ethylene glycol and frozen in liquid N2.

The complex DCAF1-**15** were obtained by incubation of cleaved His6-TEV-DCAF1(1039–1401)Q1250L at 1.6 mg/mL with a 1:200 dilution of Trypsin Gold (Promega) for 2 h on ice. The cleaved protein was further purified on HiLoad 16/60 Superdex 75 equilibrated with 50 mM HEPES pH 7.5 200 mM NaCl, 0.5 mM TCEP. Fractions containing a fragment of ~27 kDa were pooled and concentrated to 13 mg/mL. Protein was then mixed with 2 mM compound and incubated briefly on ice before mixing with reservoir solution 1:1 (100 nL + 100 nL) in sitting drop crystallization plates at 20 °C. Crystals were obtained within 1–3 days with reservoir solutions containing 2.13 M LiSO$_4$ and 0.1 M Tris pH 7.5 and were cryo-protected by quickly transferring the crystals to reservoir solution containing additional 20%(V/V) ethylene glycol.

Datasets were collected at the PXII beamline at the Swiss Lightsource). Data were integrated by XDS[80] and successively merged and scaled by AIMLESS[81] in the CCP4I2 interface[82]. Molecular replacement was performed using Phaser[83] with the starting coordinates 4PXW. Refinement was performed in iterative cycles of modelbuilding in Coot[84] and refinement in Refmac5[85]. Geometrical correctness of the model was validated by Molprobity[86]. Coordinates were deposited in the PDB with accession codes: 8OO5 and 8OOD and data processing and refinement statistics can be found in Supplementary Table 1.

## Chemical proteomics

**Synthesis of azide-functionalized resin.** Azide-modified sepharose fastflow4 resin was created by adding 2 μmol of 11-azido-3,6,9-trioxaundecan-1-amine/mL bead volume and 30 μL triethylamine in anhydrous DMSO. This coupling reaction was allowed to proceed overnight at room temperature with end-over-end agitation. Next, 100 μL 2-(2-aminoethoxy)ethanol was added directly to the coupling reaction and allowed to proceed overnight at room temperature with end-over-end agitation. The resulting azide-functionalized resin was then washed several times with anhydrous DMSO.

**Synthesis of DCAF1 affinity matrix.** 10 μmol (**16**) was added to 1 mL of the azide-functionalized resin in the presence of 1 mM CuSO$_4$, 1 mM TBTA, and 50 mM TCEP in anhydrous DMSO. The coupling reaction was allowed to proceed overnight at room temperature with end-over-end agitation, then washed extensively with anhydrous DMSO.

**Lysate generation.** HEK293T cell pellets were suspended in lysis buffer (50 mM HEPES pH 7.5, 150 mM NaCl, 1.5 mM MgCl$_2$, 1 mM DTT, 0.8% NP-40, 5% glycerol, and 1x protease inhibitors (Thermo)) at 2× pellet volume.

The samples were sonicated on ice for 30 s at 20% amplitude with a 2 second pulse ON and 3 s pulse OFF. Cell debris was removed by centrifugation at 20,000 × $g$ for 20 min at 4 °C. Protein concentration was determined by BCA (Pierce) and diluted down to 5 mg/mL with pulldown buffer (50 mM HEPES pH 7.5, 150 mM NaCl, 1.5 mM MgCl$_2$, 1 mM DTT, 0.4% NP-40, 1× protease inhibitors (Thermo Fisher Scientific)).

**Chemical proteomics pulldown experiment.** 1 mL of 5 mg/mL HEK293T lysates was treated with 100 μM (**16**) or DMSO for 1 h at 4 °C in replicate. Following incubation, the lysates were incubated with 35 μL (**16**)-functionalized resin 4 °C for 4 hours. Unbound proteins on affinity matrix were washed and bound proteins were eluted with 2X LDS sample buffer (NuPAGE, Invitrogen) containing 10 mM DTT, alkylated with 25 mM iodoacetamide, and processed through detergent removal columns (Thermo). In-solution digestion and isobaric labeling with TMT reagents for the four samples (126/130 for (**16**); 127/131 for the DMSO) was performed according to standard procedures. Equal amounts of labeled peptides were pooled and separated by high pH reverse phase separation (RP10) using a Dionex UltiMate 3000 (HPLC) system (Thermo Fisher Scientific) and an Xbridge C18 3.5 μm 2.1 × 150 mm HPLC column (Waters). Resulting fractions were pooled to yield 12 final fractions which were dried, and reconstituted in 0.1% FA/2% acetonitrile for nLC-MS/MS analysis using a EASY nLC1200 system using a 1 cm online custom trap and 75 μm inner diameter separation column (ReproSil-Pur 120 C18-AQ, 3 μm material; Dr. Maisch GmbH; 90 min water-acetonitrile gradient containing 0.1% formic acid) coupled to an Orbitrap Fusion Lumos Tribrid mass spectrometer (Thermo Fisher Scientific). MS1 spectra were acquired at 120k resolution in the Orbitrap and MS2 spectra were acquired in higher-energy C-trap dissociation (HCD) mode at 30k resolution in the Orbitrap (isolation width 0.8, collision energy 40%).

Data analysis was performed using an in-house data analysis pipeline utilizing Mascot (Matrix Science), Transproteomic pipeline v3.3sqall (Institute for Systems Biology) and the Homo sapiens UniProt protein (downloaded 2017) database supplemented with common contaminants. Peptide mass tolerance was set to 10 ppm, fragment tolerance set to 0.1 Da and trypsin cleavage specificity allowing for two missed cleavages. Carbamidomethylation of cysteine was set as fixed modification, methionine oxidation and TMT-modification of N-termini and lysine residues were set as variable modifications. Peptide and protein data validation was performed using PeptideProphet and ProteinProphet modules in TPP and high confidence protein identifications were reported using a false positive prediction of <1% by ProteinProphet. Peptide quantitation was achieved using the sum of TMT reporter ion intensities for spectral matches to a given peptide sequence and only peptide-to-spectrum matches that were unique assignments to a given identified protein within the total dataset were considered for protein quantitation. Protein fold changes were calculated as median peptide fold change and $p$ values were calculated using a one-way $T$-test (arbitrarily set to 1 for non-significant single peptide quantitation) and adjusted using the Benjamini-Hochberg False Discovery Rate (FDR). Only proteins with two or more quantified peptides are reported in the final dataset. The dataset has been uploaded to the PRIDE database with the accession code PXD047347.

## Immunoblotting

Whole cell lysates for immunoblotting were prepared by pelleting cells at 4 °C (300 g) for 3 min. Cell pellets were then washed 1x with PBS and resuspended in RIPA lysis buffer (VWR, cat # 97063-270) supplemented with protease (Thermo Scientific, cat # A32955) and phosphatase (Thermo Scientific, cat # A32957) inhibitor tablets. Lysates were clarified at 16,000 × $g$ for 15 min at 4 °C prior to quantification by Lowry assay (Bio-Rad cat # 5000113 and cat # 5000114). Whole-cell lysates were loaded into 4–20% Criterion TGX Precast 18 well gels (Bio-Rad, cat # 5671094) and separated by electrophoreses at 120 V for 1 h. The gels

were transferred to a nitrocellulose membrane using the Trans-Blot Turbo (Bio-Rad) for 7 min and then blocked for 1 h at room temperature in Odyssey blocking buffer (LICOR Biosciences, cat # 927-50000). Membranes were probed with the appropriate primary antibodies (diluted 1:1000) overnight at 4 °C in 20% Odyssey blocking buffer in 1x TBST. Membranes were washed three times with 1x TBST (5 min per wash), and then incubated with IRDye goat anti-mouse (LICOR, cat # 926-32210) or goat anti-rabbit (LICOR, cat # 926-32211) secondary antibody diluted 1:10,000 in 20% Odyssey blocking buffer for 1 h at room temperature. After three 5-min washes with 1x TBST, the immunoblots were visualized using the ODYSSEY Infrared Imaging System (LICOR). Alternatively, SDS-PAGE resolved proteins were transferred to a PVDF membrane using a wet blotting system (Invitrogen). Membranes were blocked in 5% Milk-PBS-Tween 20 for 1 h and incubated with primary antibody over night at 4 °C. Secondary antibodies (secondary antibody: anti-rabbit, goat pAb to Rb-IgG (Abcam, #7090); secondary antibody: anti-mouse, sheep IgG HRP-linked (GE healthcare, #NA931V)) were incubated for 1 h at room temperature (RT) before blots were developed on film using the detection kit SuperSignal West Femto Maximum Sensitivity Substrate (Thermo Fisher, #34096).

For capillary western, $4 \times 10^4$ 786-O cells were plated per well in a 96 well plate overnight, treated with compound for 4 h, cells were washed 1X with PBS, lysed with 40 µl RIPA lysis buffer (VWR, cat # 97063-270) supplemented with protease (Thermo Scientific, cat # A32955) and phosphatase (Thermo Scientific, cat # A32957) inhibitor tablets, and lysates were clarified by centrifugation at $3000 \times g$ for 45 min at 4 °C. Lysates were not diluted and samples were prepared using the standard method provided by the ProteinSimple Jess capillary western blot instrument and run using the 66-440 kDa Fluorescence Separation Kit (SM-FL005) with primary antibodies (BRD9 Bethyl A303-781A, and α-tubulin Abcam AB7291) diluted to 1:200 and HRP-conjugated mouse (Bio-techne #DM-002) and rabbit (Bio-techne #DM-001) secondary antibodies. Protein peaks were analyzed using Compass for Simple Western version 5.0.1 (ProteinSimple), area under the curve for both α-tubulin (30 kDa) and BRD9 (90 kDa) per capillary was extracted and BRD9 area was normalized to α-tubulin area on a per-capillary basis before each sample was normalized to the DMSO for each run of three triplicates.

Uncropped images of the blots can be found in the Source Data File.

### Engineering of BRD9-HiBiT/FF/CAS9 cells

HEK293/BRD9-HiBiT pools (PROMEGA # CS3023120) and HEK293/BRD7-HiBiT pools (PROMEGA # CS3023124) were purchased from Promega. We performed clonal selection for HEK293/BRD9-HiBiT cell line using single-cell sorting and screening for HiBiT expression using Nano-Glo HiBiT Lytic Detection System (Promega # N3030). The HEK293/BRD9-HiBiT clone and HEK293/BRD7-HiBiT pool to perform standard HiBiT assays. To perform well-based CRISPR, we further engineered the HEK293/BRD9-HiBiT clone by lentiviral delivery of both Cas9 in pNGx-LV-c004 as described previously[87], and FireFly Luciferase (in pNGx-LV-u003, constructed in-house, using an EF1alpha promoter, C-terminal PEST domain fusion, and a hygromycin resistance cassette). To produce lentiviral particles, each plasmid was co-transfected with ViraPower™ Lentiviral Packaging Mix into 293T cells. Supernatant from both plasmid transfections containing viral particles were spun down at $1500 \times g$ for 10 min, aliquoted and frozen at −80 °C. Thawed virus supernatant was used to co-infect 293 BRD9-HiBiT cells. And selected for Cas9 with 10 µg/ml Blasticidine, and for Firefly luciferase (FF) with 300 µg/ml Hygromycin for two weeks to obtain BRD9-HiBiT/CAS9/FF expressing cell lines. We performed clonal selection from this BRD9-HiBiT/CAS9/FF pool by single cell sorting; multiple recovering clones were then tested for HiBiT and FF signal using Nano-Glo HiBiT Dual-Luciferase Reporter System (Promega # CS1956A08) as well as CAS9 expression by western blotting using CAS9 antibody (Cell Signalling Technology cat# 14697).

All subsequent experiments described in this publication were performed with this isolated clone referred to as BRD9-HiBiT/CAS9/FF cells.

### BRD9-HiBiT protein abundance luminescence assay

To measure BRD7 and BRD9 abundance in the BRD7-HiBiT pool, BRD9-HiBiT clone, and BRD9-HiBiT/FF/cas9 cells, pre-treated cells were treated with corresponding ubiquitination pathway inhibitors (**NAE1i**, **26Si** or **UAE1i**), as indicated, for 2 h before **DBr-1** PROTAC treatment at various doses for 2 h. HiBIT (and FF, in the case of BRD9-HiBiT/FF/cas9 cells) signals were measured using either the Nano-Glo HiBiT Lytic assay (Promega #N3040) or Nano-Glo HiBiT Dual-Luciferase Reporter System (Promega # CS1956A08) according to the manufacturer's protocol with an EnVision plate reader (Perkin Elmer).

### sgRNA-mediated control knockout studies

Individually designed sgRNAs targeting *BRD9* (a mix of three guides: CGACTTTGATCCTGGGAAGA, GCCTCTAAAGCTAGTCCTGA, GTCACG GATGCAATTGCTCC), *DCAF1* (mix A: TCACACATCTTGAACCTTCC and GAAGGAACCCTTACCCTGGA; mix B: CACGCGACACTGATCCAGAG and TCGCGTGGTGTTCAATCACA; mix C: GATGTTCCGTTTCACAT CAA and AGTCTTTGGTGTCTGTACAC), *PSMA1* (mix of 4 guides: ACA AGACGAGACACAGGCAG, ATGACAATGATGTCACTGTT, CCGCAATTG AGATACCAATA, and TTTTATGCGTCAGGAGTGTT), *CUL4A* (individual guides, guide A: CATAGGTGCGGTCCAAGAAC, guide B: CGCAGTTG CTTGTAGAGCAT, guide C: CTCCTCGAGGTTGTACCTGA, and guide D: TCTCGCGCTCGATCAGCAGT), *CUL4B* (individual guides, guide A: CATCAAACCCTACAAACTCC, guide B: GCCGAATCCCTGGGTTGTAA, guide C: GCTTCTTCTGTATCGGTACG, guide D: GTTTGATGCGAAGAT GGCTG), *DDB1* (individual guides, guide A: CAATAATGCCGGT CTCTGAG, guide B: GACATCATTACGCGAGCCCA, guide C: GATCT ATGTGGTCACCGCCG, guide D: GTCGTACAACTACGTGGTAA), *RBX1* (individual guides, guide A: ATGGATGTGGATACCCCGAG, guide B: GAAGAGTGTACTGTCGCATG, guide C: GTCCATTGGACAACAGAGAG, guide D: TCGCTGGCTCAAAACACGAC) and *sgAAV1* as non-targeting ctrl (GGGGCCACTAGGGACAGGAT) were cloned into pNGx-LV-g003 lentiviral plasmid as previously described[87] for lentiviral infection of BRD9-HiBiT/CAS9/FF cells. Lentiviral particles were produced as described above and cells were infected either with sgRNA-expressing particles against an individual target gene or co-infected with a combination of two particles against *CUL4A* and *CUL4B* (volume ratio 1:1) and selected 24 h post infection for 4 days with 1 µg/ml Puromycin. To determine genetic rescue from BRD9 PROTAC mediated degradation we performed a BRD9-HiBiT protein abundance luminescence assay as described.

### Arrayed CRISPR screen with an ubiquitin pathway sgRNA sublibrary

The arrayed ubiquitin pathway rescue screen is outlined in Fig. 4a. On day 1, we seeded 500 BRD9-HiBiT/FF/CAS9 293 cells per well in 384 well plates (Corning Cat # 3570) and stamped on day 2 each library plate in triplicate assay plates. Per assay plate well, we used 1 µl of ready-to-use lentiviral particles of the ThermoFisher Ubiquitin CRISPR LentiArray Library (A42270) targeting 943 genes with 3722 sgRNAs, that are pooled by four guides per gene in average. On day 3, infected cells were selected with 1 µg/ml Puromycin. On day 7, after 4 consecutive days of Puromycin selection, we incubated cells for 2 h with BRD9 PROTAC **DBr-1** before performing BRD9-HiBiT and FireFly luminescence assay using the Nano-Glo HiBiT dual-Luciferase Reporter System (Promega # CS1956A08) according to the manufacturer's manual.

### Analysis of arrayed CRISPR screen

For hit calling, we set a threshold based on a minimal Firefly signal > log10(1.25), this threshold was based on signal measurements of media containing as well as non-infected, puromycin selected control wells. The Firefly signal served as control for well-based variation of cell

number and was used to normalize the BRD9-HiBiT signals on a plate basis using a non-linear correlation (Supplementary Fig. 5c, d) from this normalized data. We calculated a robust z-score per gene (Supplementary Figure. 5e) to feed into a gene-level RSA statistical model[51] to calculate gene effect scores as max. rel. change and significance (Fig. 4d).

### Proteomics profiling of DCAF1 PROTACs

HEK293T and TMD8 cells were cultured according to protocol and 3 million cells were plated per sample and recovered for 24 h. The following day, cells were treated with indicated compounds at corresponding concentrations for 6 h. Cells were then harvested and washed 2× with PBS before removing all liquid and snap freezing in liquid nitrogen before sample preparation for proteomics.

TMT-labeled peptides were generated with the iST-NHS kit (PreOmics) and TMT10plex (duplicates per condition) or TMT18plex (triplicates per condition) reagent (Thermo Fisher Scientific). Equal amounts of labeled peptides were pooled and separated on a high pH fractionation system with a water-acetonitrile gradient containing 10 mM ammonium formate, pH 11[88]. Alternating rows of the resulting 72 fractions were pooled into 24 samples, dried and resuspended in water containing 0.1% formic acid.

The LC-MS analysis was carried out on an EASY-nLC 1200 system coupled to an Orbitrap Fusion Lumos Tribrid mass spectrometer (Thermo Fisher Scientific). Peptides were separated over 115 min or 180 min with a water-acetonitrile gradient containing 0.1% formic acid on a 25 cm long Aurora Series UHPLC column (Ion Opticks) with 75 μm inner diameter. MS1 spectra were acquired at 120k resolution in the Orbitrap, MS2 spectra were acquired after CID activation in the ion trap and MS3 spectra were acquired after HCD activation with a synchronous precursor selection approach[89] using 5 or 8 notches and 60k or 50k resolution in the Orbitrap.

LC-MS raw files were analyzed with Proteome Discoverer 2.4 (Thermo Fisher Scientific). Briefly, spectra were searched with Sequest HT against the Homo sapiens UniProt protein database and common contaminants (2019, 21494 entries). The database search criteria included: 10 ppm precursor mass tolerance, 0.6 Da fragment mass tolerance, maximum three missed cleavage sites, dynamic modification of 15.995 Da for methionines, static modifications of 113.084 Da for cysteines and 229.163 Da or 304.207 Da for peptide N-termini and lysines. The Percolator algorithm was applied to the Sequest HT results. The peptide false discovery rate was set to 1% and the protein false discovery rate was set to 2%. TMT reporter ions of the MS3 spectra were integrated with a 20 ppm tolerance and the reporter ion intensities were used for quantification[90]. The mass spectrometry proteomics data have been deposited to the ProteomeXchange Consortium (http://proteomecentral.proteomexchange.org) via the PRIDE partner repository[90] with the dataset identifier PXD046286. Protein relative quantification was performed using an in-house developed R (v.3.6) script. This analysis included multiple steps; global data normalization by equalizing the total reporter ion intensities across all channels, summation of reporter ion intensities per protein and channel, calculation of protein abundance log2 fold changes (L2FC) and testing (two-sided test) for differential abundance using moderated t-statistics[91] where the resulting p-values reflect the probability of detecting a given L2FC across sample conditions by chance alone. To control the False Discovery Rate (FDR), the *p* values were adjusted for multiple testing using the Benjamini-Hochberg method (referred to as *q* values). A R markdown file describing the differential abundance analysis is available on GitHub [https://github.com/Novartis/px_tmt_daa].

### Engineering of stably expressing TMD8 BTK-GFP-chysel-mCHERRY sensor cells and flow cytometry to monitor BTK-GFP degradation

The bicistronic pLenti6 BTK-GFP-chysel-mCHERRY expression vector was cloned as described[92] using Gateway cloning with a pENTR-BTK plasmid lacking a STOP codon. TMD8 cells were then infected with lentiviral particles from this vector as described above and stably transfected cells were selected with blasticidin at a concentration of 10 μg/mL referred to as TMD8 BTK-GFP/mCh cells.

For flow cytometry measurements, 25,000 exponentially growing TMD8 BTK-GFP/mCh cells were plated in 100 μL of each well of a 96-well plate and grown for 24 h. Cells were then dosed with corresponding compounds and doses from 10 mM DMSO stock for 24 h, after which GFP and mCherry fluorescence was read with a Beckmann Coulter Cytoflex. For competition experiment, both compounds were added at the same time and dosed for 24 h. Cells were gated based on Forward & Side Scatter and selection for Single cells using SSC-A vs. SSC-H. Further, the population of viable cells was defined by the forward & side-scatter characteristics of DMSO-treated cells. To confirm cells outside of this population as non-viable, cells were plated and treated for 24 h, washed with PBS and stained for 30 min with two markers of early (Annexin-V) and late (LIVE/DEAD) apoptosis. Effects on viability were measured by the reduction of the percentage of cells included in this viable-population gate with increasing doses of compound tested. Degradation of BTK was calculated based on subtracting background (BG) fluorescence of parental TMD8 signal via (GFP-BG)/(mCherry-BG) of viable cells. Median ratios of three technical (different wells) replicates were used to calculate daily averages. Each compound dose–response was generated via three biological replicates (different days of experiment). Degradation curve parameters (DC50 & Abs. DC50 & max. Deg) were extracted using GraphPad Prism's 4-parameter logistic curve fitting.

The viability window was determined by measuring the GI50 of viability based on the percentage of cells included in the viable gate and comparing it to the DC50 of degradation. The fold-change difference describing this window is [GI50 Viability / DC50 Degradation], where values above 1 indicate a positive (lower DC50 degradation than AC50 viability) window. If a compound did not have a DC50 or GI50, the top dose assessed was used for calculations.

### DCAF1 TR-FRET assay in standard binary format and cooperative format with BTK

Competition assay conditions consist of 20 μL total volume in white 384-well plates (Greiner), in 30 mM phosphate buffer pH 7.4 containing 140 mM NaCl, 0.01% Tween20, 1 mM TCEP and 1% final DMSO. Tested compounds at 14 different concentrations were pre-incubated at 22 °C for 30 min with a large excess of 10 μM human BTK (G389-S659) or 10 μM human BRD9-BD(130-250) in the cooperative assay or with buffer in standard binary assay. The following reagent was then added: 2 nM human DCAF1(1073-1399)_E1398S-Avi-His (biotinylated at the C-terminus), 0.75 nM cryptate terbium streptavidin (Perkin Elmer) and 10 nM bodipy labeled ligand (SMILES: CC1 = CC(C) = C2C = C3C = CC(CCC(N4CCN(C5 = CC6 = NC(C(C) (C7 = CC = C(C = C7)Cl)C) = NC(NCCN) = C6C = C5)CC4) = O) = N3 = B(F)(N12)F) (internal preparation, $K_D = 69 \pm 6$ nM towards DCAF1 measured by fluorescence polarization). Samples were then incubated at 22 °C for 30 min before reading using a Tecan infinite M-1000 or PheraStar FSX (excitation 340 nm, emission 490 and 520 nm). DCAF1 protein was omitted in the negative control. IC50 fitting was carried out using an in-house developed software (Novartis Helios software application, unpublished, a publicly available alternative would be e.g., Graphpad Prism) using the methods described by Fomenko et al.[93] (regression algorithms for nonlinear dose-response curve fitting). Following normalization of activity values for the wells to % inhibition (% inhibition = [(high control-sample)/(high control-low control)] × 100). Data analysis could also be carried out using GraphPad Prism without prior normalization. The reported IC50 values are the average of at least two independent experiments.

## DCAF1 BTK ubiquitination assay

To measure DCAF1 and PROTAC-dependent BTK ubiquitination activity in vitro, we reconstituted the ubiquitination system using in-house produced and commercial proteins. The assay buffer consisted of 50 mM Tris-HCl pH 7.4, 30 mM NaCl, 10 mM MgCl$_2$, 0.2 mM CaCl$_2$, 0.01% TritonX, 1 mM DTT, 20 U/mL IPP, and 0.05 mM ATP. The biochemical assay was performed in 10 μl volume in the following order of addition into the MTP: (1) 50 nl of test compound in 100% DMSO. (2) 950 nl of assay buffer. (3) 3 μl of BTK-Tb-SA pre-mixture. (4) 3 μl of E1-E2-Fluo-Ub pre-mixture. (5) 3 μl of CRL4-DCAF1. The final concentrations of components in assay were: 100 nM BTK, 4 nM Tb-SA, 100 nM E1, 2.5 μM E2, 2.5 μM Fluro-Ub, and 50 nM CRL4-DCAF1. After addition of all components (in the order listed above), the mixture was read in kinetic mode with a PheraStar fluorescence reader. in 2-minutes intervals using a standard TR-FRET protocol (LanthaScreen Optic module, Flash lamp Excitation source, 30 flashes per well).

## BTK EPK assay

BTK kinase domain was mixed in assay buffer containing 50 mM HEPES pH 7.5, 0.02% Tween20, 0.02 % bovine serum albumin, 1 mM DTT, 0.01 mM Na$_3$VO$_4$, 10 mM b-Glycerolphosphate, 18 mM MgCl$_2$, 1 mM MnCl$_2$ to a 2× concentration of 8 nM. This solution was added to pre-dispensed compound dilution series in black, 384 well -low dead volume plates (Greiner) and the reaction was initiated by the addition of ATP and a BTK consensus peptide (Sequence: 5-Fluo-Ahx-TSELKKVVALYDY-Nle-P-Nle-NAND-NH2) to a final assay concentrations of 0.047 mM and 0.002 mM, respectively in assay buffer. The reaction was stopped after 60 min at 32 °C by addition of the same volume of spot solution containing 100 mM HEPES pH 7.5, 5% DMSO, 10 mM EDTA, 0.015% Brij 35. Stopped reactions were analyzed on a Caliper LC3000, which was treated with the commercially available coating reagents CR−3 and CR-8 as instructed by the manufacturer. The relative intensities of phosphorylated peptide were calculated. Data was normalized to control wells containing 100 mM EDTA solution instead of the 2× kinase solutions (0%) and DMSO-only control wells (100%). Normalized data was fitted using a sigmoidal fit.

## Intracellular BTK engagement NanoBRET

Apparent binding affinity of test compounds for BTK in intact and permeabilized cells was assessed under steady-state (end point) competition conditions by measuring their concentration-dependent reduction of the bioluminescence resonance energy transfer signal between BTK-NanoLuc (light donor) and Tracer-05 (Promega), a fluorescent BTK ATP-site binder (light acceptor), after 2 h concurrent incubation, basically as described[94] with minor adaptations. Thus, HEK293A cells were batch-transfected by adding 1 μg pLenti6 BTK-Nanoluc vector and 9 μg Transfection Carrier DNA (pcDNA3.1 Hygro +) to 1 mL of Opti-MEM without phenol red (Thermo Fisher). After addition of 40 μL ViaFect (Promega) and incubation for 20 min, the DNA:lipid mixture was added to trypsinized HEK293A cells suspended at 2 × 10$^5$/mL in 10 ml growth medium. 100 μL of the resulting cell:DNA:lipid suspension was then seeded at 2 × 10$^4$ cells/well into white, clear-bottom 96-well plates (Costar) that were pre-coated with poly-D-Lysine by incubation for 60 min with 30 μL of a sterile-filtered solution of 0.1 mg PDL (Mpbio) dissolved in phosphate-buffered saline, followed by a wash with 200 μL sterile phosphate-buffered and airdrying. After a 20 h incubation in a humidified, 37 °C/5% CO$_2$ incubator, the cell supernatant was replaced with 95 μL serum- and phenol red-free Opti-MEM. Cells were then incubated for 2 hrs with serially diluted compounds added using an HP300 non-contact dispenser (Tecan), and 5 uL Tracer-05 (Vasta et al. 2018) pre-diluted in NanoBRET Dilution Buffer (12.5 mM HEPES, 31.25% PEG-300, pH 7.5) to achieve a final concentration of 0.5 μM. To measure BRET,

NanoBRET NanoGlo substrate (Promega, final dilution of 1:1000) and Extracellular NanoLuc Inhibitor (Promega, final concentration 20 μM) were added, followed by quantification of donor and acceptor luminescence emission on a Pherastar FX (BMG Labtech) multimode reader using 460 nm band-pass and 610 nm long-pass filters, respectively. To calculate raw BRET ratio values, the acceptor emission value was divided by the donor emission value for each sample. For background correction, the BRET ratio generated in the absence of tracer was subtracted from the BRET ratio of each sample. Raw BRET units were then converted to milliBRET units (mBU) by multiplying each raw BRET value by 1000.

The mBU values for each treated sample was then divided by the average mBU value determined based on four DMSO-treated vehicle control samples and multiplied by 100 to arrive at % of vehicle-treated control. Four-parametric dose-response curve fits (model 203, non-fixed minima and maxima) were performed using the XLfit add-in for Excel (IDBS) to determine the compound concentration achieving half-maximal tracer displacement (IC50).

To enable parallel assessment of cellular and cell-free target engagement, all compounds were tested in the absence and presence of 50 μg/mL digitonin (Sigma), respectively.

## Generation of CRBN-BTK PROTAC resistant TMD8 cells for DCAF1-BTK PROTAC rescue studies

TMD8 cells were grown in the presence of a CRBN-BTK PROTAC **CBt** using a gradual dose increase. Start treatment was 0.1 nM for 6 weeks, followed by 3 weeks at 0.4 nM and another 4 weeks at 10 nM to finally obtain **CBt** resistant TMD8 cells (TMD8 CRBN resist.).

## Statistics and reproducibility

Western plots showing degradation of POIs in this study were performed once unless otherwise indicated and their results were confirmed by orthogonal methods such as quantitative proteomics. Figure 1 depicts the range of DEMETER2 scores across WD40 containing E3 ligase proteins. Sample sizes represented by these scores range from a maximum 711 (DDB1) to a minimum 343 (FBXW12) cell line counts out of a maximum possible 712 cell lines. The complete list of these gene-sample counts is summarized in the Source data file as count. The corresponding statistical descriptors for Fig. 1a, c are collected in the source data file Fig. 1a and Fig. 1c, respectively. For all remaining experiments, the number of replicates, error bars, and statistical significance are defined in the relevant figure legends or in the respective supplementary Data files, tables or in source data file.

## Reporting summary

Further information on research design is available in the Nature Portfolio Reporting Summary linked to this article.

## Data availability

Source data are provided with this paper as a Source Data file. The full list of E3 ligase with WDR motifs is attached as Supplementary Data files 1. The raw data for the proteomics study is attached as Supplementary Data files 2, 6, and 8. The raw data for the arrayed CRISPR rescue screen is attached as Supplementary Data file 4. All compound smiles can be found in Supplementary Data files 9. The coordinates and structure factors of the DCAF1-**13** and DCAF1-**15** complexes have been uploaded to the PDB (PDB IDs: 8OO5 and 8OOD). All proteomic data have been uploaded to PRIDE under the identifiers PXD046286 and PXD047347. Source data are provided with this paper.

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

## Acknowledgements

We thank our extended team members, the TPD initiative and in particular Marcel Reck, Alexandre Luneau, Emine Sager, Corinne Marx, B. Forrester, Anne-Sophie Mangold, and C. Wiesmann for support and scientific input, and the NIBR Innovation Postdoctoral program for support of M.S. We also thank Prof. S. Tohda for providing TMD8 cells.

## Author contributions

Conceptualization of the manuscript is performed by M.S., M.R., C.R.T., T.R., P.H., A.V., N.H.T. The original draft was written by M.S., M.R., C.R.T. and further revised/edited by T.R., M.R., A.H., M.S., L.T., G.H., C.R.T. The following authors contributed experimentally to the manuscript: M.S., M.R., X.L., F.M., S.F., F.G., X.Li, M.-T.S., J.T., D.B., P.L., R.A.-R., S.C., B.P., A.H., M.Schi, D.G., K.C., B.B.-P., M.M., M.N., R.M., M.H., J.A., M.K. The chemistry was performed by T.Z., N.S., B.Y.C., R.M., P.I. and the data and computational analysis by F.S., S.G., and E.A. Work in Thomä laboratory was supported by funding from the European Research Council (ERC) under the European Union's Horizon 2020 Research and Innovation Program grant agreement no. 666068, European Research Council (ERC) under the European Union's H2020 research program (NucEM, no. 884331), SNF 31003A_179541, CRSII5_186230 and 310030_301206, from Krebsforschung (KFS 4980-02-2020), the Gebert Rüf Stiftung (GRS-057/14) and the Novartis Research Foundation.

## Competing interests

M.S., X.L., F.M., T.Z., S.F., F.G., X.Li, F.S., S.G., T.-M.S., J.T., D.B., P.L., R.A.-R., B.Y.C., S.C., B.P., A.H., M.Schi, N.S., D.G., K.C., B.B.-P., M.M., M.N., R.M., M.H., J.A., E.A., G.H., L.T., A.V., M.K. are employees and shareholders of Novartis Pharma. M.R., C.R.T., T.R., R.M., and P.I. are former employees of Novartis. N.H.T. receives funding from the Novartis Research Foundation and is a scientific advisory board (SAB) member of Monte Rosa Therapeutics and an advisor to Zenith Therapeutics and Ridgeline. The remaining authors declare no competing interests.

## Additional information

[1]Novartis Institutes for BioMedical Research, Basel, Switzerland. [2]Novartis Institutes for BioMedical Research, Cambridge, MA, USA. [3]Friedrich Miescher Institute for Biomedical Research, Basel, Switzerland. [4]Present address: Ridgeline Discovery, Basel, Switzerland. [5]Present address: Swiss Institute for Experimental Cancer Research (ISREC), École Polytechnique Fédérale de Lausanne, Lausanne, Switzerland. [6]These authors contributed equally: Martin Schröder, Martin Renatus. ✉e-mail: martin-1.schroeder@novartis.com; cthoma@ridgelinediscovery.com

