## [Peer Review File · Nature Communications]

Reviewers' Comments:

Reviewer #1:

Remarks to the Author:

Review of Schröder et al.

Discovering new E3 ligase ligands for PROTACs is an important and timely endeavour of the field of targeted protein degradation that is necessary to expand the chemical and biological space beyond CRBN and VHL. Successful outcome in this direction will derisk and expand the scope of protein degraders as a therapeutic modality that today is witnessing almost 30 PROTACs molecules in clinical trials.

Here a team at NIBR report on their discovery of novel DCAF1 small molecule ligands and qualify their use a novel E3 handle in PROTAC design. The motivation and rationale for choosing DCAF1 was two-fold. First, it is the substrate recognition subunit of the Cullin RING ligase 4 (CRL4) complex, and is an essential gene in cancer cells, hence could raise the bar and so potentially delay onset of resistance to DCAF1-based PROTACs in future clinical settings. Second, because DCAF1 contains a highly ligandable WD40 repeat domain novel ligands could be developed with high-binding affinity (~10-100 nM of Kd), specificity (as shown in chemoproteomics pull-down experiments) and with crystallographically-defined binding mode. The data presented here on these ligands is solid and convincing although only minimal data to evidence the key ligands is presented. The authors state that they will disclose the full journey to the discovery and SAR of these compounds in an accompanying publication in future, which seems reasonable, so I feel this amount of data is sufficient here.

They then qualify utility of such ligands as DCAF1 handles in the design and characterization of active PROTAC degraders, using three different proteins of interest and POI ligands as model systems: 1) Brd9, 2) dasatinib and related TKI targets, and BTK. Convincing evidence is provided to establish what is now considered as gold-standard to qualify a useful E3 ligase handle in the field, in a manner consistent and aligned with what the field has been demonstrating since 2015 over and over again for multiple targets with CRBN and VHL based ligands and PROTACs. Namely, the crucial points of evidence provided here are: a) that the degradation activity is dependent on DCAF1; b) that their PROTAC degraders have the expected mode of action i.e. via ternary complex formation / ubiquitination. C) that utility of a DCAF1 based PROTAC degrader could be established in a CRBN-resistance setting.

Given the broad target scope and the evidence-base provided as summarized above, this study in my opinion offers an important and compelling discovery that warrants publication in Nature Communication. This advance could have important impact to the field. It could usher a novel E3 ligase that is more essential than CRBN and VHL. This perhaps in future could expand the clinical development of PROTACs. The work also establishes DCAF1 ligase and ligands as new tools for TPD. The future will tell if these ligands will become as popular as the "top two", CRBN and VHL, and the data presented here is definitely promising.

There are really many other positives to this study. The protein structural biology and biophysics of binary/ternary complexes is very well done, and so the cell biology proteomics and chemistry. However, there are also several negatives and limitations to this study. First, the lack of presentation of a suitable negative control at the E3 ligase, which is an important requirement for any new E3 handles in the field, as utility of this has been widely demonstrated with VHL (cis-Hyp) and CRBN (methylation at the glutarimide ring). I feel this is a significant limitation of this study and should be addressed prior to publication. Another weakness of the manuscript in my opinion is the part on the SAR of the BTK PROTACs (Figure 4). I found the data on this sparse and of limited value to the story, and as a result the discussion on this part is rather convoluted and confusing and the overall picture emerges as fuzzy and inconclusive. I am particularly unconvinced about the whole part attempting to rationalize SAR for relating permeability/TC formation/ubiquitination/degradation of the BTK PROTAC series as the data presented in Figure 4 is sparse and insufficient with only limited compounds. It is well known in the field that PROTAC design and SAR is multi factorial and lot of steps are at play and there can be different kinetic regimes so I do not think this part adds much to the story or to the field of PROTAC design per se,

and instead I believe that it may actually obfuscate and diminish the prominence of the main finding / message of the paper. I would therefore strongly encourage the authors to substantially revise / remove this part. Extensive discussions on SAR would appear more appropriate to follow-on med chem papers.

There are other important aspects that require the authors attention and consideration that in my opinion would improve the manuscript and strengthen its impact to the field, as described below. I do not mean to diminish the high-quality and soundness of the work and of the data presented, which would eventually warrant publication. However I feel there is significant scope to improve the paper as is, including new experimentation to address the concern raised as well as revising the narrative of the manuscript to ensure the key findings and take home messages are clearly delivered and not mixed up. To this end, I hereby offer the authors further comments and thoughts for their consideration.

Major points and scope to strengthen the paper with inclusion of new experimentation:

1. The data on Brd9 PROTACs is OK but as the first target POC presented I feel there is scope to strengthen this part. It is well established that the bromodomain ligand used engages not just Brd9 but also Brd7. Furthermore, CRBN and VHL based PROTACs are both available (dBRD9 and VZ185, respectively) and two CRBN PROTAC degraders are in the clinic. Interestingly the CRBN based PROTACs degrade highly selectively Brd9, whereas VZ185 is a dual Brd9/7 degrader (albeit with slight preference for Brd9. All of the above observations suggest that it would seem important and warranted that the authors duly consider a) to include monitoring of Brd7 protein levels with the DCAF1 PROTACs cellular data presented; b) to compare and benchmark their degraders vs dBRD9 and VZ185 head to head.

2. In my opinion, one of the most important lines of evidence to support the claim and raise impact fo this work is that presented in the final Figure, that goes towards showing the advantage or differentiation of their DCAF1 PROTACs in settings where CRBN or VHL PROTACs are limited or resistance has emerged. Here, the data presented in Figure 5 showing degradation and cellular cytotoxicity of their DCAF1 BTK PROTAC in a CRBN-BTK PROTAC resistant setting is the critical data and expanding this data seems warranted. Have the authors thought of including similar evidence in settings where VHL based PROTACs are ineffective e.g. in RCC4 cells lacking VHL or cell lines resistant to VHL-based PROTACs? How would a DCAF1 based PROTAC compare with CRBN/VHL PROTACs in a wider cell panels sensitivity screen e.g. those performed by OncoLead (see Ottis et al. ACS Chem Biol 2019, PMID: 31553577). Some new data along these or other lines would strengthen this important claim and line of evidence, augmenting the impact of this study and its relevance to the field.

Other points

Could the authors comment on :

- The importance of the free terminal amino group in the SAR and E3 ligase binding. If this is strictly required, could the authors comment on any implication in drug development and potential PK liabilities
- Half-life of DCAF1 (is it known from literature? have the team measured this?)
- Are the endogenous substrate(s) of DCAF1 known and if so it would be interesting and important to monitor their protein levels upon PROTAC treatments.
- Feasibility of a DCAF1 Knock-Out cell line, which would be an ideal tool to confirm the tightness of the DCAF1 dependencies of the compounds induced protein degradation. Presumably hard to attain due to the essentiality of DCAF1 in cells? Please comment. Note that imo the data presented complaining sgRNA data (Fig. 2G) and CRISPR knock-out screen (Fig. H) provide unambiguous evidence and are sufficient for the purpose of establishing the CRL4-DCAF1 dependency of the MoA.
- The conjugation away of the DCAF1 ligand, albeit chemically-intuitive given the suitability of the Piperazinyl-amide group as conjugation site, appears to offer an exit vector that points towards the side of the beta-propeller fold. Achieving an optimal geometry to position the recruited target protein towards the ubiquitination zone of the CRL4 complex might therefore require some twist of the linker back towards the centre of the substrate recognition domain, similar to the structure of

MZ1 (and in a manner consistent with the model they present in the Supp Figure 1 based on their cocrystal structure with compound 15). Have the authors considered performing any modeling of the ternary complexes formed by their most active PROTACs. This is not a requirement for publication, yet a figure showing how a putative complex e.g. with the BRd9 bromodomain might geometrically position in the context of the whole CRL4 complex to enable effective ubiquitination would be interesting to the audience (see related modeling from Gadd et al. 2017 Nat Chem Biol for CRL2-VHL MZ1 Brd4-BD, and Bai et al J Biol Chem 2022 PMID: 35101445 for CRBN PROTACs) - The readers would be most interested in learning more about the authors' thought process about their choices of targets to validate utility of their DCAF1 ligands in PROTACs. The most widely-used POI system in TPD to test and establish new E3 ligase ligands has arguably been to date JQ1/BET proteins Brd2/3/4. I am curious if the authors tried it and what was observed. If they did, any particular reasons the authors have not included data on this? I should pre-empt here that I am not at all advocating for including data on Brd4/JQ1, and as explained with three target/ligand systems presented that does more than suffice.

Other points of attention on Figures:

Figure 2 SPR data (panel B) - it would be interesting if the authors could set up their experiments in such a way as to extract K_{off} (hence $t_{1/2}$) as well as cooperativity of the ternary complex when they titrate and flow the saturated binary Brd9:PROTAC over the immobilized DCAF1 (like they have done with BTK PROTACs?). This would help the reader understand the level of thermodynamics and kinetics properties of their system and make comparisons with those in the literature for the archetypical BET:MZ1:VHL system and other systems studied in the same way

Competition experiments (Fig 2 E and Fig 5 C) - it would be interesting and important to include competition with the respective Brd9 and DCAF1 binding ligands in degradation experiments which is gold-standard in PROTAC MOA studies.

Figure 4B - could include cooperativity α values in the table

I really like Figure 4 panel C overlaying bell-shaped PROTAC titration curves for biophysical SPR ternary complex formation assays over Ubiquitination rates of DCAF1 measured in ubiquitin-transfer biochemical assays. I do not believe to have seen this done before in the field but it makes a very visual understanding of how ternary complex formation relates to ubiquitination. Warrants further description or at least a note to highlight this point, imo. The authors might want to show the raw ubiquitination data in one of the main figure panels, rather than just plotting the quantification.

Figure 5D: what are the two bands in DCAF1 blot, and why does the bottom band disappear in the samples in lanes 2, 4, 6 and 8 (numbering from left to right?)

There are also parts of the manuscript text that require attention:

Pg. 5: "has been shown to be amenable for PROTAC-mediated degradation through CRBN (Remillard et al., 2017)." The authors might want to mention this has been achieved via VHL too (our PROTAC VZ185) and cite the related paper (Zoppi et al. J Med Chem 2019)

Bottom of pg. 9: "Design of compounds with a high degree of cooperativity and prolonged ternary complex half-life has recently been suggested as a promising strategy to enhance PROTAC efficacy." Respectfully, I feel that this sentence should really include citations to work from our lab that has shown this in particular to the paper where this conceptual advance was first proposed (Gadd et al. Nat Chem Biol 2017) and to one of the first example of applying such strategy purposely in PROTAC optimization (Farnaby et al. Nat Chem Biol 2019).

Discussion:

"However, further comparative studies are needed to better understand substrate preferences of DCAF1 compared to other E3 ligase receptors such as CRBN or VHL." I am not sure I follow the logic of the authors. Since this sentence follows a discussion on the impact of proteins cellular

localization to the observed effectiveness of induced protein degradation, I was expecting at this point more a statement along the lines of: "However, further comparative studies are needed to better understand how the relative cellular localizations of DCAF1 and recruited substrates might impact on how effectively they might be degraded by DCAF1 PROTACs."

As mentioned above, I found the data on SAR of BTK PROTACs sparse and of limited value to the story, and the discussion on this part rather confusing and inconclusive. Specifically, in the discussion, the long paragraph: "Our systematic discovery approach led to the potent DCAF1-BTK PROTAC DBt-10 and highlighted the successful change of a prototypic PEG linker-based PROTAC to a more complex drug-like molecule. " reads as rather convoluted and leaves the reader more confused than persuaded. As mentioned above, I am really unconvinced about this whole part attempting to rationalize SAR for relating permeability/TC formation/ubiquitination/degradation of the BTK PROTAC series as the data is sparse and insufficient. I do not think this part adds much to the story and main finding / message of the paper, and would fit better into a follow-up SAR paper e.g. in a more specialized Med Chem journal

Similarly, the subsequent paragraph:

"It is important to note that all the biophysical and biochemical studies were performed with truncated BTK (kinase domain) and DCAF1 (WDR domain) and the cellular experiments were all performed with full-length BTK and endogenous DCAF1 that might confound some of the observations, especially in the light of the complex activation mechanisms reported for DCAF1." As mentioned, I feel this whole part does not add much to this story

"Further studies are necessary to understand the full potential of DCAF1 as an efficient ligase for PROTAC-mediated degradation, and our limited comparative studies between DCAF1 and CRBN PROTACs (dasatinib: DDa-1 vs CDa-1, Fig. 3B; BTK: DBt-10 vs CBt, Fig. 5D) revealed that in both cases CRBN-mediated degradation was more efficient, although this is beyond the scope of this work. "

Here, I feel the authors have not given themselves a chance to properly benchmark and compare side by side their DCAF1 based PROTACs with CRBN and on this regard also VHL based PROTACs. This should be relatively straightforward to do for Brd9 PROTACs including dBRD9 and VZ185, and both compounds are available from vendors. For BTK, the authors might want to include a VHL based PROTAC too.

Pg. 18: the labels GQN626 and IVH258 come out of the blue. Presumably they refer to the DCAF1 ligands? Please define what they are or used consistent labeling of compounds throughout . THE same applies for VHF543 (legend of Figure 1d and methods)

Supp Fig 4 d : legend describing the SPR data is incomplete

Supporting INformation document seems to be missing a Reference list.

I agree to waive my anonymity as referee in a spirit to enhance the transparency and constructiveness of the peer review process, Alessio Ciulli

Reviewer #2:

Remarks to the Author:

This paper by Schröder et al. presents a remarkable study on the discovery of a potent binder for the E3 ligase DCAF1, with implications for targeted protein degradation. The authors build upon previous findings of ligands for the WDR domain protein EED and focus specifically on DCAF1, an essential E3 ligase receptor of the CRL4 subfamily. This selection is motivated by the potential of DCAF1, an essential gene, to overcome drug-resistance issues associated with non-essential E3 ligases.

The authors successfully identify a potent binder for DCAF1 that occupies the WDR donut-hole pocket. They thoroughly confirm the binding affinity and specificity of the binder using various techniques such as surface plasmon resonance (SPR), X-ray crystallography, and

chemoproteomics. Furthermore, by linking the DCAF1 binder to different ligands that target various proteins including tyrosine kinases, BRD9, and BTK, the authors demonstrate the ability to degrade a diverse range of proteins of interest by harnessing DCAF1 and its binder. Notably, the DCAF1-based BTK PROTAC even exhibits degradation activity in a cell line that is resistant to CRBN-based degraders.

The experimental design in the paper is commendable, encompassing a CRISPR KO rescue screen, whole proteome analysis, structural analysis, as well as cellular and binding assays. These well-designed experiments provide robust support for the findings presented. Overall, this research paper offers valuable insights into the discovery and characterization of a selective DCAF1 E3 ligase binder. The implications of this work extend to the development of non-covalent heterobifunctional PROTACs capable of degrading diverse disease-causing proteins. Thus, this paper represents a significant contribution to the field of protein degradation and is well-suited for Nature Communications.

I have a minor question regarding the paper. It would be good if the authors could provide some clarification. In Figure 5D, it is observed that DCAF1 exhibits two distinct bands. However, upon treatment with the degrader, it seems that the lower band disappears. Could the authors speculate on the nature of this lower band?

Dear Dr. Ossa and reviewers,

Thank you very much for your positive and constructive feedback. Please find below a point-by-point response of your comments. Our responses are highlighted in blue.

Review of Schröder et al.

Discovering new E3 ligase ligands for PROTACs is an important and timely endeavour of the field of targeted protein degradation that is necessary to expand the chemical and biological space beyond CRBN and VHL. Successful outcome in this direction will derisk and expand the scope of protein degraders as a therapeutic modality that today is witnessing almost 30 PROTACs molecules in clinical trials.

Here a team at NIBR report on their discovery of novel DCAF1 small molecule ligands and qualify their use a novel E3 handle in PROTAC design. The motivation and rationale for choosing DCAF1 was two-fold. First, it is the substrate recognition subunit of the Cullin RING ligase 4 (CRL4) complex, and is an essential gene in cancer cells, hence could raise the bar and so potentially delay onset of resistance to DCAF1-based PROTACs in future clinical settings. Second, because DCAF1 contains a highly ligandable WD40 repeat domain novel ligands could be developed with high- binding affinity (~10-100 nM of Kd), specificity (as shown in chemoproteomics pull-down experiments) and with crystallographically-defined binding mode. The data presented here on these ligands is solid and convincing although only minimal data to evidence the key ligands is presented. The authors state that they will disclose the full journey to the discovery and SAR of these compounds in an accompanying publication in future, which seems reasonable, so I feel this amount of data is sufficient here.

They then qualify utility of such ligands as DCAF1 handles in the design and characterization of active PROTAC degraders, using three different proteins of interest and POI ligands as model systems: 1) Brd9, 2) dasatinib and related TKI targets, and BTK.

Convincing evidence is provided to establish what is now considered as gold-standard to qualify a useful E3 ligase handle in the field, in a manner consistent and aligned with what the field has been demonstrating since 2015 over and over again for multiple targets with CRBN and VHL based ligands and PROTACs. Namely, the crucial points of evidence provided here are: a) that the degradation activity is dependent on DCAF1; b) that their PROTAC degraders have the expected mode of action i.e. via ternary complex formation / ubiquitination. c) that utility of a DCAF1 based PROTAC degrader could be established in a CRBN-resistance setting.

Given the broad target scope and the evidence-base provided as summarized above, this study in my opinion offers an important and compelling discovery that warrants publication in Nature Communication. This advance could have important impact to the field. It could usher a novel E3 ligase that is more essential than CRBN and VHL. This perhaps in future could expand the clinical development of PROTACs. The work also establishes DCAF1 ligase and ligands as new tools for TPD. The future will tell is these ligands will become as popular as the “top two”, CRBN and VHL, and the data presented here is definitely promising.

There are really many other positives to this study. The protein structural biology and biophysics of binary/ternary complexes is very well done, and so the cell biology proteomics and chemistry. However, there are also several negatives and limitations to this study. First, the lack of presentation of a suitable negative control at the E3 ligase, which is an important requirement for any new E3 handles in the field, as utility of this has been widely demonstrated with VHL (cis-Hyp) and CRBN (methylation at the

glutarimide ring). I feel this is a significant limitation of this study and should be addressed prior to publication.

We thank Reviewer 1 for his constructive feedback. We acknowledged the lack of the negative control and we addressed this concern experimentally by producing compound 13-N (new characterization data in Figure 1 and Supplementary figure 1) by di-methylation of the primary amine of 13 (Synthesis described in the supplementary information). Compound 13-N showed in TR-FRET and SPR experiments a >100x fold reduction in affinity for DCAF1 while retaining similar chemical properties. Additionally, we synthesized the corresponding control of the DCAF1-BRD9 degrader DBr-1, named DBr-1-N, which showed 27x reduced DCAF1 affinity compared to DBr-1 (Supplementary Figure 3). Of note, the not-so-reduced affinity of the final BRD9 degrader control PROTAC DBr-1-N compared to the ligand 13-N (0.812uM compared to >17uM in TR-FRET assays) was also reflected by a residual weak degradation of BRD9 confirmed by western blot analysis and HiBiT assays (Supplementary Figure 3). We also observed a high degree of cooperativity for DBr-1-N potentially caused by DCAF1-BRD9 direct interactions (See Figure 2, and Supplementary Figure 3). We could confirm that this remaining BRD9 degradation activity is due to DCAF1 engagement since co-treatment with both compound 13 and OICR-8268 rescued DBr-1N mediated BRD9 degradation in HiBiT assays (Supplementary Figure 4). Thus, although our newly generated negative control maintains some degree of activity towards the POI, we believe it still serves its purpose, as supported by the validation experiments described above, and we hope the referee will appreciate this new data.

Another weakness of the manuscript in my opinion is the part on the SAR of the BTK PROTACs (Figure 4). I found the data on this sparse and of limited value to the story, and as a result the discussion on this part is rather convoluted and confusing and the overall picture emerges as fuzzy and inconclusive. I am particularly unconvinced about the whole part attempting to rationalize SAR for relating permeability/TC formation/ubiquitination/degradation of the BTK PROTAC series as the data presented in Figure 4 is sparse and insufficient with only limited compounds. It is well known in the field that PROTAC design and SAR is multi factorial and lot of steps are at play and there can be different kinetic regimes so I do not think this part adds much to the story or to the field of PROTAC design per se, and instead I believe that it may actually obfuscate and diminish the prominence of the main finding / message of the paper. I would therefore strongly encourage the authors to substantially revise / remove this part. Extensive discussions on SAR would appear more appropriate to follow-on med chem papers.

We thank reviewer 1 for this feedback and substantially revised this part. We reduced the number of BTK PROTACs described here to the two best compounds, DBt-5 and DBt-10, showed their in-depth characterization and focus, due to its better degradation -to-toxicity ratio, on DBt-10 (Figure 6, Supplementary Figures 7-9). We additionally collected proteomics data for this compound in TMD8 cells highlighting its high selectivity for BTK (only one other kinase, LIMK2, was also degraded, but to a lesser extend). Further, we rewrote the discussion in accordance with the reviewer's suggestions. We hope that this more focused and concise description of our BTK-PROTACs efforts will be clearer to the readers and less distracting of the main message of DCAF1 as a novel ligase.

There are other important aspects that require the authors attention and consideration that in my opinion would improve the manuscript and strengthen its impact to the field, as described below.

I do not mean to diminish the high-quality and soundness of the work and of the data presented, which would eventually warrant publication. However I feel there is significant scope to improve the paper as is, including new experimentation to address the concern raised as well as revising the narrative of the manuscript to ensure the key findings and take home messages are clearly delivered and not mixed up. To this end, I hereby offer the authors further comments and thoughts for their consideration.

Major points and scope to strengthen the paper with inclusion of new experimentation:

1. The data on Brd9 PROTACs is OK but as the first target POC presented I feel there is scope to

strengthen this part. It is well established that the bromodomain ligand used engages not just Brd9 but also Brd7. Furthermore, CRBN and VHL based PROTACs are both available (dBRD9 and VZ185, respectively) and two CRBN PROTAC degraders are in the clinic. Interestingly the CRBN based PROTACs degrade highly selectively Brd9, whereas VZ185 is a dual Brd9/7 degrader (albeit with slight preference for Brd9. All of the above observations suggest that it would seem important and warranted that the authors duly consider a) to include monitoring of Brd7 protein levels with the DCAF1 PROTACs cellular data presented; b) to compare and benchmark their degraders vs dBRD9 and VZ185 head to head.

We thank reviewer 1 for those two comments. We performed new experiments based on these suggestions and compared dBRD9, VZ185 and DBr-1 side by side in orthogonal assays to measure BRD7 degradation in addition to BRD9 (see new data in Figure 3) This data show that our DCAF1 based degrader is, unlike dBRD9, partially degrading BRD7, however to a much lesser extent than the dual degrader, VZ185. We hope that this direct comparison further highlights the opportunities for DCAF1 as a novel ligase with potential for slightly different substrate preference than VHL and CRBN.

2. In my opinion, one of the most important lines of evidence to support the claim and raise impact for this work is that presented in the final Figure, that goes towards showing the advantage or differentiation of their DCAF1 PROTACs in settings where CRBN or VHL PROTACs are limited or resistance has emerged. Here, the data presented in Figure 5 showing degradation and cellular cytotoxicity of their DCAF1 BTK PROTAC in a CRBN-BTK PROTAC resistant setting is the critical data and expanding this data seems warranted. Have the authors thought of including similar evidence in settings where VHL based PROTACs are ineffective e.g. in RCC4 cells lacking VHL or cell lines resistant to VHL-based PROTACs? How would a DCAF1 based PROTAC compare with CRBN/VHL PROTACs in a wider cell panels sensitivity screen e.g. those performed by OncoLead (see Ottis et al. ACS Chem Biol 2019, PMID: 31553577). Some new data along these or other lines would strengthen this important claim and line of evidence, augmenting the impact of this study and its relevance to the field.

We followed the suggestion by reviewer 1 and performed a comparative study in renal cell carcinoma cells, using 768-O cells as a representative model, which lack VHL expression and thus represent a system intrinsically resistant to VHL-based PROTACs. Using the same three BRD9 PROTACs (dBRD9, VZ185 and DBr-1) we could confirm that the DCAF1 exploiting PROTAC, DBr-1, was able to degrade BRD9 in a context where VHL-based PROTACs are ineffective. We believe this demonstrates that DCAF1 PROTACs offer a viable alternative to VHL-based PROTACs in such indications (new data in Figure 7). In the light of this new findings we adapted the title of the manuscript to: "Generation of selective and reversible DCAF1-based PROTACs with cellular activity against clinically validated targets and in intrinsic- and acquired-degrader resistant settings".

Other points

Could the authors comment on :

- The importance of the free terminal amino group in the SAR and E3 ligase binding. If this is strictly required, could the authors comment on any implication in drug development and potential PK liabilities

We thank reviewer 1 for this comment. We tested the di-methylated analogue of compound 13, named 13-N, and due to its dramatic loss of affinity used this as a potential negative control compound. (see new characterization data as mentioned above in Figure 1 and Supplementary Figure 1) Therefore, we concluded that substitution the primary amine, as undesirable as it might be for further drug development, could impair the affinity towards DCAF1.

- Half-life of DCAF1 (is it known from literature? have the team measured this?)

We followed up on this comment and found that the half-life of DCAF1 in TMD8 cells has been described previously to be greater than 20h for 5 different immune system related cell lines, by Mathieson et al. (Mathieson et al., 2018). Additionally, internal data by SILAC experiments places the half-life around 8-11h for the cell-lines used in this manuscript (see the figure below for referees).

- Are the endogenous substrate(s) of DCAF1 known and if so it would be interesting and important to monitor their protein levels upon PROTAC treatments.

We thank reviewer 1 for this comment. There are few studies describing endogenous substrates for DCAF1. A recent preprint proposes PLK4 as a substrate of DCAF1 (Grossmann et al., 2023). However, the authors of this preprint mapped the binding site to the unstructured C-terminal region beyond the WD40 domain and we therefore did not test the effects of our compounds on PLK4 level. Nevertheless, we believe that our compounds could be a great resource to evaluate in further studies the substrate spectrum of DCAF1.

- Feasibility of a DCAF1 Knock-Out cell line, which would be an ideal tool to confirm the tightness of the DCAF1 dependencies of the compounds induced protein degradation. Presumably hard to attain due to the essentiality of DCAF1 in cells? Please comment. Note that imo the data presented complaining sgRNA data (Fig. 2G) and CRISPR knock-out screen (Fig. H) provide unambiguous evidence and are sufficient for the purpose of establishing the CRL4-DCAF1 dependency of the MoA.

We thank the reviewer for this comment. In our hands the creation of a stable DCAF1 KO cell-line was not possible, most likely due its high essentiality.

- The conjugation away of the DCAF1 ligand, albeit chemically-intuitive given the suitability of the Piperazinyl-amide group as conjugation site, appears to offer an exit vector that points towards the side of the beta-propeller fold. Achieving an optimal geometry to position the recruited target protein towards the ubiquitination zone of the CRL4 complex might therefore require some twist of the linker back towards the centre of the substrate recognition domain, similar to the structure of MZ1 (and in a manner consistent with the model they present in the Supp Figure 1 based on their cocrystal structure with compound 15). Have the authors considered performing any modeling of the ternary complexes formed by their most active PROTACs. This is not a requirement for publication, yet a figure showing how a putative complex e.g. with the BRd9 bromodomain might geometrically position in the context of the whole CRL4 complex to enable effective ubiquitination would be interesting to the audience (see related modeling from Gadd et al. 2017 Nat Chem Biol for CRL2-VHL MZ1 Brd4-BD, and Bai et al J Biol Chem 2022 PMID: 35101445 for CRBN PROTACs)

We followed the helpful suggestion of reviewer 1 and modeled the potential complex with DDB1, Cul4A and RBX1 and the highlighted potential ubiquitination zone (new data in Supplementary Figure 1D). We hope that our here described compounds spark further structural work which could help to determine the structure of the ternary complex and will enable rational linker design to increase the cooperativity.

- The readers would be most interested in learning more about the authors' thought process about their choices of targets to validate utility of their DCAF1 ligands in PROTACs. The most widely-used POI system in TPD to test and establish new E3 ligase ligands has arguably been to date JQ1/BET proteins Brd2/3/4. I am curious if the authors tried it and what was observed. If they did, any particular reasons the authors have not included data on this? I should pre-empt here that I am not at all advocating for including data on Brd4/JQ1, and as explained with three target/ligand systems presented that does more than suffice.

In this study we did not focus on the BET family and JQ1 and have not tested if JQ1-fusions with our ligands would lead to BRD2/3/4 degradation. We hope that the targeted protein degradation community will use the here described approach to develop such degraders and test them for BET degradation and selectivity within this family.

Other points of attention on Figures:

Figure 2 SPR data (panel B) - it would be interesting if the authors could set up their experiments in such as way as to extract K_{off} (hence $t_{1/2}$) as well as cooperativity of the ternary complex when they titrate and flow the saturated binary Brd9:PROTAC over the immobilized DCAF1 (like they have done with BTK PROTACs?). This would help the reader understand the level of thermodynamics and kinetics properties of their system and make comparisons with those in the literature for the archetypical BET:MZ1:VHL system and other systems studied in the same way

We performed such a SPR experiment as well as TR-FRET assays (at two tracer concentrations) in presence and absence of additional BRD9 protein (Figure 2 and Supplementary Figure 2). We also performed such experiments for the control degrader DBr-1-N (Supplementary Figure 3).

Competition experiments (Fig 2 E and Fig 5 C) - it would be interesting and important to include competition with the respective Brd9 and DCAF1 binding ligands in degradation experiments which is gold-standard in PROTAC MOA studies.

We thank reviewer for this suggestion. We followed the advice and tested our BRD9 degrader DBr-1 and the BTK PROTAC DBt-10 in such competition assays. For DBr-1 we observed that the DCAF1 ligands compound 13 and OICR-8268 did not rescue BRD9 degradation at the used concentrations (Supplementary Fig. 4). We speculated that the high degree of cooperativity between the two proteins BRD9 and DCAF1 impairs the competition capacity of the DCAF1 ligand only. Nevertheless, 10 μ M of the BRD9 ligand BI-9564 were able to rescue the DBr-1 mediated degradation. We also performed the same experiment for the control degrader DBr-1-N, which has weaker recruitment capacity (described in detail above, data in Supplementary Figure 4). Here we observed full rescue of BRD9 degradation with both DCAF1 BRD9 ligands.

We did a similar experiment in TMD8 cells with the BTK degrader DBt-10 using the BTK-GFP/mCherry reporter assay described in this study. This assay demonstrated not only a dose-dependent rescue effect of various DCAF1 and BTK ligands, but also was able to serve as a FACS-based cellular target engagement assay (see new data in Supplementary figure 9). Since this PROTAC system demonstrates lower cooperativity than DBr-1, we were able to observe a higher rescue efficiency with DCAF1 ligands .

Figure 4B - could include cooperativity α values in the table

Following the advice to reshape this entire section we redesigned the entire figure (See Figure 6).

I really like Figure 4 panel C overlaying bell-shaped PROTAC titration curves for biophysical SPR ternary complex formation assays over Ubiquitination rates of DCAF1 measured in ubiquitin- transfer biochemical

assays. I do not believe to have seen this done before in the field but it makes a very visual understanding of how ternary complex formation relates to ubiquitination. Warrants further description or at least a note to highlight this point, imo. The authors might want to show the raw ubiquitination data in one of the main figure panels, rather than just plotting the quantification.

We thank reviewer 1 for the positive feedback. We incorporated the same comparison also for DBt-10 and blotted the raw initial velocities in Supplementary Figure 7.

Figure 5D: what are the two bands in DCAF1 blot, and why does the bottom band disappear in the samples in lanes 2, 4, 6 and 8 (numbering from left to right?)

We thank reviewer 1 for this comment. We have observed inconsistencies in the expression pattern of the lower band across many different western blots in various cell lines. Based on shRNA validation, we believe the upper band is DCAF1 because we did not observe loss of the lower band with two of three shRNAs (shown below). Since this non-specific band can lead to confusion, we have cropped the image to only show the top band (Figure 7C) but, along with other westerns, have the full-length blot in the

source data.

There are also parts of the manuscript text that require attention:

.

Pg. 5: "has been shown to be amenable for PROTAC-mediated degradation through CRBN (Remillard et al., 2017). " The authors might want to mention this has been achieved via VHL too (our PROTAC VZ185) and cite the related paper (Zoppi et al. J Med Chem 2019)

We thank reviewer 1 for pointing this out and added the citation of this work.

Bottom of pg. 9: "Design of compounds with a high degree of cooperativity and prolonged ternary complex half-life has recently been suggested as a promising strategy to enhance PROTAC efficacy." Respectfully, I feel that this sentence should really include citations to work from our lab that has shown this in particular to the paper where this conceptual advance was first proposed (Gadd et al. Nat Chem Biol 2017) and to one of the first example of applying such strategy purposely in PROTAC optimization (Farnaby et al. Nat Chem Biol 2019).

We agree with reviewer 1, but following the advice given before regarding the BTK-SAR part we removed this entire paragraph.

Discussion:

“However, further comparative studies are needed to better understand substrate preferences of DCAF1 compared to other E3 ligase receptors such as CRBN or VHL. “ I am not sure I follow the logic of the authors. Since this sentence follows a discussion on the impact of proteins cellular localization to the observed effectiveness of induced protein degradation, I was expecting at this point more a statement along the lines of:“However, further comparative studies are needed to better understand how the relative cellular localizations of DCAF1 and recruited substrates might impact on how effectively they might be degraded by DCAF1 PROTACs.”

We thank the reviewer for this helpful comment. We rephrased this paragraph to reflect better the sub-cellular location and substrate preference.

As mentioned above, I found the data on SAR of BTK PROTACs sparse and of limited value to the story, and the discussion on this part rather confusing and inconclusive. Specifically, in the discussion, the long paragraph: “Our systematic discovery approach led to the potent DCAF1-BTK PROTAC DBt-10 and highlighted the successful change of a prototypic PEG linker-based PROTAC to a more complex drug-like molecule. “ reads as rather convoluted and leaves the reader more confused than persuaded. As mentioned above, I am really unconvinced about this whole part attempting to rationalize SAR for relating permeability/TC formation/ubiquitination/degradation of the BTK PROTAC series as the data is sparse and insufficient. I do not think this part adds much to the story and main finding / message of the paper, and would fit better into a follow-up SAR paper e.g. in a more specialized Med Chem journal

We thank reviewer 1 for this feedback. As mentioned above we removed this discussion part and substantially revised the result paragraph describing the BTK PROTACs. By removing most of the data and focusing only on the characterization data of two DCAF1-BTK PROTACs (DBt-5 and DBt-10) we hope that the manuscript focuses better on the main message of the use of reversible DCAF1 ligands to develop DCAF1-PROTACs for different targets and settings resistance to existing PROTACs.

Similarly, the subsequent paragraph:

“It is important to note that all the biophysical and biochemical studies were performed with truncated BTK (kinase domain) and DCAF1 (WDR domain) and the cellular experiments were all performed with full-length BTK and endogenous DCAF1 that might confound some of the observations, especially in the light of the complex activation mechanisms reported for DCAF1.”

As mentioned, I feel this whole part does not add much to this story

As mentioned above we have removed this section.

“Further studies are necessary to understand the full potential of DCAF1 as an efficient ligase for PROTAC-mediated degradation, and our limited comparative studies between DCAF1 and CRBN PROTACs (dasatinib: DDa-1 vs CDa-1, Fig. 3B; BTK: DBt-10 vs CBt, Fig. 5D) revealed that in both cases CRBN-mediated degradation was more efficient, although this is beyond the scope of this work. “. Here, I feel the authors have not given themselves a chance to properly benchmark and compare side by side their DCAF1 based PROTACs with CRBN and on this regard also VHL based PROTACs. This should be relatively straightforward to do for Brd9 PROTACs including dBRD9 and VZ185, and both compounds are available from vendors. For BTK, the authors might want to include a VHL based PROTAC too.

We thank reviewer 1 for the constructive and helpful comment. As mentioned above, we performed the suggested comparison of dBRD9, VZ185 and DBr-1 (new data in Figure 3). Following the advice we also removed this statement from the discussion and rephrased it taking into account the new data, which suggest that DCAF1 could serve as alternative E3 ligase in addition to CRBN and VHL.

Pg. 18: the labels GQN626 and IVH258 come out of the blue. Presumably they refer to the DCAF1 ligands? Please define what they are or used consistent labeling of compounds throughout. The same applies for VHF543 (legend of Figure 1d and methods)

We thank reviewer 1 for pointing this out and replaced the compound descriptions with the ones used in this manuscript.

Supp Fig 4 d : legend describing the SPR data is incomplete

We are thankful for highlighting this. We revised the supplementary figures and the captions based on all feedback provided by the reviewers and hope to present a more complete supplementary information file.

Supporting Information document seems to be missing a Reference list.

We thank reviewer 1 for this comment We added the reference list in supplementary Information.

I agree to waive my anonymity as referee in a spirit to enhance the transparency and constructiveness of the peer review process, Alessio Ciulli

Reviewer #2 (Remarks to the Author):

This paper by Schröder et al. presents a remarkable study on the discovery of a potent binder for the E3 ligase DCAF1, with implications for targeted protein degradation. The authors build upon previous findings of ligands for the WDR domain protein EED and focus specifically on DCAF1, an essential E3 ligase receptor of the CRL4 subfamily. This selection is motivated by the potential of DCAF1, an essential gene, to overcome drug-resistance issues associated with non-essential E3 ligases.

The authors successfully identify a potent binder for DCAF1 that occupies the WDR donut-hole pocket. They thoroughly confirm the binding affinity and specificity of the binder using various techniques such as surface plasmon resonance (SPR), X-ray crystallography, and chemoproteomics. Furthermore, by linking the DCAF1 binder to different ligands that target various proteins including tyrosine kinases, BRD9, and BTK, the authors demonstrate the ability to degrade a diverse range of proteins of interest by harnessing DCAF1 and its binder. Notably, the DCAF1-based BTK PROTAC even exhibits degradation activity in a cell line that is resistant to CRBN-based degraders.

The experimental design in the paper is commendable, encompassing a CRISPR KO rescue screen, whole proteome analysis, structural analysis, as well as cellular and binding assays. These well-designed experiments provide robust support for the findings presented. Overall, this research paper offers valuable insights into the discovery and characterization of a selective DCAF1 E3 ligase binder. The implications of this work extend to the development of non-covalent heterobifunctional PROTACs capable of degrading diverse disease-causing proteins. Thus, this paper represents a significant contribution to the field of protein degradation and is well-suited for Nature Communications.

I have a minor question regarding the paper. It would be good if the authors could provide some clarification. In Figure 5D, it is observed that DCAF1 exhibits two distinct bands. However, upon treatment with the degrader, it seems that the lower band disappears. Could the authors speculate on the nature of

this lower band?

We thank reviewer 1 and reviewer 2 for their comments. We have observed inconsistencies in the expression pattern of the lower band across many different western blots in various cell lines. Based on shRNA validation, we believe the upper band is DCAF1 because we did not observe loss of the lower band with two of three shRNAs (shown below). Since this non-specific band can lead to confusion, we have cropped the image to only show the top band (Figure 7C) but, along with other westerns, have the

full-length blot in the source data.

- Grossmann, J., Kratz, A.-S., Kordonsky, A., Prag, G., & Hoffmann, I. (2023). The CUL4-DDB1-DCAF1 E3 ubiquitin ligase complex regulates PLK4 protein levels to prevent premature centriole duplication. *bioRxiv*, 2023.2009.2013.555514. <https://doi.org/10.1101/2023.09.13.555514>
- Mathieson, T., Franken, H., Kosinski, J., Kurzawa, N., Zinn, N., Sweetman, G., Poeckel, D., Ratnu, V. S., Schramm, M., Becher, I., Steidel, M., Noh, K.-M., Bergamini, G., Beck, M., Bantscheff, M., & Savitski, M. M. (2018). Systematic analysis of protein turnover in primary cells. *Nature Communications*, 9(1), 689. <https://doi.org/10.1038/s41467-018-03106-1>

Thank you again for all your comments and feedback.

Sincerely,

Claudio Thoma and Martin Schröder

Martin Schröder
Innovation Post-Doc
CHEMICAL BIOLOGY & THERAPEUTICS / CBT - NIBR
WSJ-182 3 102.05
CH-4056 Basel
martin-1.schroeder@novartis.com

 **NOVARTIS**

Reviewers' Comments:

Reviewer #1:

Remarks to the Author:

The authors have done an outstanding job at revising the paper taking onboard my comments and critique. They now provide extensive new data that in full address all of my concerns and as a result have substantially strengthened the paper. In particular, the new ligand with much-reduced binding affinity is included and shown utility by incorporating it in place of the high-affinity DCAF1 binding ligand into a full PROTAC (here for Brd9). This has yielded a molecular matched pair PROTAC analogue that no longer degrades the target protein at the same low concentration as the active degrader. This is excellent, and although the compound is not strictly a negative control compound, the authors provide data in support as to why that is the case (less of a reduction in binary binding and on top of that positive cooperativity within the ternary complex that together likely compensate the effect and result still in some DCAF1-dependent degradation). The revised manuscript now also includes extensive new data comparing and benchmarking their BRD9 PROTAC side by side against VZ185 (VHL) and dBRD9 (CRBN) and it fares very respectably. The assessment of their PROTAC in VHL-lacking cells is also a great addition and truly supports the revised title that speaks to addressing both intrinsic and acquired resistance settings. Finally, I commend the authors for taking on board my suggestion of significantly revisiting the BTK protac part – I feel that this now reads much more clearly and fits much better as part of the story, and does not distract as much away from the main message of the paper. Well done! The major and important new data together with the nicely revised figures and narrative have substantially improved the paper, and I now support its rapid publication in the journal.

Before the paper is ready for publication, however, the authors might want to fix a few minor outstanding issues that I have noticed while reading through the revised manuscript:

- The authors now cite Zoppi et al. J Med Chem 2019 (for VZ185 and VHL-based Brd9 PROTACs) – however the reference appears to be missing in the list of references at the end of the paper.
- Please spell out how many replicates were performed for each treatment in the whole-cell TMT-labeling MS proteomics experiments. This is important to understand the robustness of the statistical analysis.
- The main text mentioned LIMK2 as the only significantly depleted protein, in addition to BTK, however in the corresponding figure (6F) the corresponding dot is labelled as LIMK1 (while LIMK2 is labelled for a non-degraded protein)?

Congratulations again! I am sure the whole TPD field will look forward to learn of this exciting discovery once published soon.

Alessio Ciulli

Reviewer #2:

Remarks to the Author:

The authors did a great job revising the manuscript and have addressed all my concerns. I recommend publication of this manuscript in its current form without reservations.

Dear Cara and reviewers,

Thank you very much for your positive and constructive feedback. Please find below a point-by-point response of your comments. Our responses are highlighted in blue.

REVIEWERS' COMMENTS

Reviewer #1 (Remarks to the Author):

The authors have done an outstanding job at revising the paper taking onboard my comments and critique. They now provide extensive new data that in full address all of my concerns and as a result have substantially strengthened the paper. In particular, the new ligand with much-reduced binding affinity is included and shown utility by incorporating it in place of the high-affinity DCAF1 binding ligand into a full PROTAC (here for Brd9). This has yielded a molecular matched pair PROTAC analogue that no longer degrades the target protein at the same low concentration as the active degrader. This is excellent, and although the compound is not strictly a negative control compound, the authors provide data in support as to why that is the case (less of a reduction in binary binding and on top of that positive cooperativity within the ternary complex that together likely compensate the effect and result still in some DCAF1-dependent degradation). The revised manuscript now also includes extensive new data comparing and benchmarking their BRD9 PROTAC side by side against VZ185 (VHL) and dBRD9 (CRBN) and it fares very respectably. The assessment of their PROTAC in VHL-lacking cells is also a great addition and truly supports the revised title that speaks to addressing both intrinsic and acquired resistance settings. Finally, I commend the authors for taking on board my suggestion of significantly revisiting the BTK protac part – I feel that this now reads much more clearly and fits much better as part of the story, and does not distract as much away from the main message of the paper. Well done! The major and important new data together with the nicely revised figures and narrative have substantially improved the paper, and I now support its rapid publication in the journal.

Before the paper is ready for publication, however, the authors might want to fix a few minor outstanding issues that I have noticed while reading through the revised manuscript:

- The authors now cite Zoppi et al. J Med Chem 2019 (for VZ185 and VHL-based Brd9 PROTACs) – however the reference appears to be missing in the list of references at the end of the paper.

Thank you very much for spotting this. We updated the list of references

- Please spell out how many replicates were performed for each treatment in the whole-cell TMT-labeling MS proteomics experiments. This is important to understand the robustness of the statistical analysis.

We added the number of replicates in the method section and figure captions and uploaded all raw data to EBI PRIDE (ID: PXD046286 (DOI 10.6019/PXD046286)).

- The main text mentioned LIMK2 as the only significantly depleted protein, in addition to BTK,

however in the corresponding figure (6F) the corresponding dot is labelled as LIMK1 (while LIMK2 is labelled for a non-degraded protein)?

Thank you very much for spotting this. We corrected this typo to LIMK1.

Congratulations again! I am sure the whole TPD field will look forward to learn of this exciting discovery once published soon.

Alessio Ciulli

Reviewer #2 (Remarks to the Author):

The authors did a great job revising the manuscript and have addressed all my concerns. I recommend publication of this manuscript in its current form without reservations.

We thank for Reviewer 2 for their positive feedback.

Thank you again for all your comments and feedback.

Sincerely,

Claudio Thoma and Martin Schröder

Martin Schröder

Innovation Post-Doc

DISCOVERY SCIENCES / DSc - BR

WSJ-182 3 102.05

CH-4056 Basel

martin-1.schroeder@novartis.com

 **NOVARTIS**